# A Pattern Language for Machine Learning Tasks

## Abstract

We formalise the essential data of objective functions as equality constraints on composites of learners. We call these constraints "tasks", and we investigate the idealised view that such tasks determine model behaviours. We develop a flowchart-like graphical mathematics for tasks that allows us to; (1) design and optimise desired behaviours model-agnostically; (2) offer a unified perspective of approaches in machine learning across domains; (3) import insights from theoretical computer science into practical machine learning. As a proof-of-concept of the potential practical impact of our theoretical framework, we exhibit and implement a novel "manipulator" task that minimally edits input data to have a desired attribute. Our model-agnostic approach achieves this without the need for custom architectures, adversarial training, random sampling, or interventions on the data, hence enabling capable, small-scale, and training-stable models.

## 1 Introduction

The primary instrument for controlling the training of machine learning (ML) models is the objective function, which can be broken into three modular parts. Let $\Theta_e, \Theta_d$ be the parameter-spaces of a model enc and dec respectively. Then the reconstruction loss that characterises an autoencoding task amounts to minimising (for $\theta_e \in \Theta_e, \theta_d \in \Theta_d$) the following objective function:

$$\text{argmin}_{\theta_e, \theta_d} \left( \mathbb{E}_{x \sim \mathcal{X}}[\mathbf{D}(\text{dec}_{\theta_d}(\text{enc}_{\theta_e}(x)), x)] \right)$$

We take the expected value over a data distribution $\mathcal{X}$ of a measure of statistical divergence $\mathbf{D}$ (such as cross-entropy or log-likelihood) of two expressions that, under ideal conditions, should be equal: the decoding of the encoding of some data $x$, and the original $x$. In this work, we focus on the two expressions that want to be equal, and we call this equational constraint a *task*.

In practice, designing a good objective function incorporates many technical choices, such as choice of architecture, measure of statistical divergence, and training data (Ciampiconi et al., 2023; Richardson, 2022; Terven et al., 2023). However, these choices are often made by heuristics, or rationalised post hoc. While such choices are sometimes required to make training tractable, they are not always relevant to understanding the final behaviour of the trained model. Instead, we propose and investigate the idealised view that:

**Tasks determine model behaviour.**

**To reason about tasks compactly, we use flowchart-like string diagram notation**.

**Example 1.1.** For a hyperparameter choice of divergence $\mathbf{D}$, where $X$ is an input datatype, and $LAT$ is the datatype of the latent space, the two components of the empirical risk minimisation of the autoencoder *task* consist of (1) applying the encoder enc (typed $X \to LAT$) followed by the decoder dec (typed $LAT \to X$) to inputs $x$ drawn from a source of data $\mathcal{X}$ over the datatype $X$, *which should be equal to* (2) the original $x$. We depict this as:

$$\mathbb{E}_{x \sim \mathcal{X}}[\mathbf{D}(\underbrace{dec_{\theta_d}(enc_{\theta_e}(x))}_{(1)}, \underbrace{x}_{(2)})] \iff x \begin{array}{c} \overset{X}{\vdash} \boxed{\text{enc}} \overset{LAT}{} \boxed{\text{dec}} \overset{X}{} \\ \underbrace{\phantom{xxxxxx}}_{dec_{\theta_d}(enc_{\theta_e}(x))} \end{array} \overset{\text{enc,dec}}{=} x \begin{array}{c} \overset{X}{\vdash} \\ \underbrace{\phantom{xx}}_{x} \end{array}$$

The diagrammatic notation is formally equivalent to the traditional symbolic notation with the addition of type-information about inputs and outputs. While the formal details (Appendix A) involve category theory, the power of string diagrams lies in their intuitive visual nature: by reading the diagrams as flowcharts from left to right, practitioners can leverage these diagrams to reason about ML tasks without needing to fully grasp the underlying mathematical formalism. In summary, nodes depicted as various shapes are functions, and wires are datatypes which can be understood as carrying information.

To explore the expressive power of tasks, **we abstract away implementation details such as architecture and training by idealising models to be** *universal function approximators* that can, in principle, perfectly optimise objective functions. Thus each task can be viewed purely as an equational constraint on the behaviour of the learners, comparable to equational constraints on the possible values of variables in algebra. **This perspective allows us two ways to** *specialise* **tasks by imposing structural inductive biases**, by specifying architectural choices, or by adding additional objectives.

**Example 1.2** (Residuation as an architectural choice). Diagrammatically, choosing an architecture means substituting a "black-box" universal function approximator with another diagram with matching input-output type constraints; intuitively, since a universal function approximator can be any function, it can *in particular* be a specific function of the same input-output type if necessary. The formal semantics for such substitutions are provided in Appendix A.2. A simple example is residuation, where a learner $N$ of type $X \to X$ is transformed into $N^{res} := x \mapsto N(x) + x$, depicted as:

**Example 1.3** (Perceptual losses as multi-objective learning). A common form of multi-objective learning is to add a normalisation or regularisation term to an objective function, for instance, a perceptual loss $\mathbf{L} : Y \to \mathbb{R}^{\geq 0}$ (which we depict with a "white box", because there are no learnable parameters). Multiple tasks may be combined into single objective functions by means of weighted summation with positive hyperparameter-coefficients $\alpha, \beta$.

$$\text{argmin}_{\theta_e, \theta_d} \left( \mathbb{E}_{x \sim \mathcal{X}}[\mathbf{D}(\text{dec}_{\theta_d}(\text{enc}_{\theta_e}(x)), x)] \right) \quad \Rightarrow \quad \begin{aligned} \text{argmin}_{\theta_e, \theta_d} \Big( & \alpha \mathbb{E}_{x \sim \mathcal{X}}[\mathbf{D}(\text{dec}_{\theta_d}(\text{enc}_{\theta_e}(x)), x)] \\ & + \beta \mathbb{E}_{x \sim \mathcal{X}}[\mathbf{D}(\mathbf{L}(\text{dec}_{\theta_d}(\text{enc}_{\theta_e}(x)), 0))] \Big) \end{aligned}$$

Specialisation of tasks obtains desirable properties without compromising basic behaviours, and allows us to explain the behaviour of models in terms of their constituent tasks. As a canonical example, we may obtain a variational autoencoder (VAE) (Kingma & Welling, 2022) from a regular one by the two kinds of specialisation described.

**Example 1.4** (VAE). Setting the output type of the encoder to be a space (mean, variance) of parameters for Gaussians; declaring the decoder to be internally structured as the sequential composite of sampling from a Gaussian of the input (mean, variance) followed by a learner; adding an additional normalisation task which requires the outputs of the encoder to be close to $(0, 1)$ — the parameters of the unit Gaussian. enc maps the input space $X$ to $(x, \sigma)$, the parameter-space of Gaussians over the latent space $LAT$. The decoder is the composite of a sampling function samp. $: (\mu, \sigma) \to LAT$ and a learner dec $: LAT \to X$. Note the additional NORMALISE task that enc must satisfy. This presentation of VAEs is equivalent to the traditional probabilistic form when the statistical divergence is KL (Rocca, 2021).

## 1.1 CONTRIBUTIONS

We make three primary contributions in this work.

Our first, theoretical contribution is **the formalisation of a common but informal standard procedure in deep learning**, which may be summarised as the following recipe:

1. Characterise desired behavior via equational constraints (tasks) between learners
2. Implement tasks by treating neural networks as universal approximators
3. Convert equations to loss function by a choice of hyperparameters, namely weighted sum of constituent losses and choice of divergences

Our second, theoretical contribution is a demonstration that **using our framework, we may analyse, predict, and design behaviours of models**. Using specialisation and algebraic reasoning, we can analyse the behaviour of complex models in terms of simple, well-understood tasks we call *patterns*. Moreover, we can define a novel synthetic relationship between tasks called *refinement* which describes when one optimally trained set of tasks entails satisfaction of a different set of tasks. Altogether, we may use these techniques to understand and compare the behaviour of models before committing to potentially costly training. Example 2.6 and Propositions 2.9, 3.2, and 3.3 illustrate the kinds of formal reasoning our language enables.

Our third, practical contribution and proof-of-concept is the **implementation of a novel problem class we call `manipulation`** (Section 3), which formalises (Bancilhon & Spyratos, 1981) the problem of viewing and editing a targeted attribute of data while "leaving other aspects the same". Notably, this task represents the first problem class to our knowledge that appears to be naturally solved by the relatively underexplored technique of "multi-learner multi-objective learning", which our framework naturally accommodates. As examples in image domains, we; change only the colour of a shape (Figure 1); change the value of a handwritten digit without affecting other stylistic properties (Figure 2); and change only whether a person is smiling in an image (Figure 6). Even in these toy domains, we observe a range of benefits we expect to scale: by following our recipe we obtain architecture-agnostic (Table 1) style-transference models without the need for randomness, adversarial training, or modality- and architecture-specific interventions, with good interpolation properties (Figure 6).

We conclude by discussing relations to similar approaches in the literature, along with avenues and prospects for further development.

## 2 Tasks and patterns

### 2.1 Tasks

We assume the following contextual data, omitted if there is no confusion. Let $X, Y$ denote datatypes; $\Sigma$ a set of processes $f$, each of which has (possibly empty) learnable parameter datatypes $\mathfrak{p}_f$. An atomic task is a process-theoretic equational constraint on learners specifying that $f$ should behave like $g$ on all inputs. The objective function of an atomic task $\varphi$ of type $X \rightarrow Y$ equipped with distribution $\mathcal{X}$ corresponds to a map $\mathfrak{p}_\varphi \rightarrow \mathbb{R}$ that sends $\pi \mapsto \mathbb{E}_{x \sim \mathcal{X}}[\mathbf{D}(\mathrm{sys}_{\varphi;\pi}(x), \mathrm{spec}_{\varphi;\pi}(x))]$ for some choice of statistical divergence $\mathbf{D}$. A learner can optimise multiple atomic tasks simultaneously by optimising a combination $\alpha$ of the atomic objective functions (commonly obtained by taking a weighted sum).

**Definition 2.1** (Tasks). An *atomic task* $\varphi$ is a tuple $(f, g, \mathcal{X}, \mathfrak{p})$, where $f, g : X \rightarrow Y$ are composite processes of $\Sigma$, $\mathcal{X}$ is a distribution over $X$, and $\mathfrak{p} \subseteq \mathfrak{p}_f \oplus \mathfrak{p}_g$ is a space of trainable parameters. We indicate the *system* $f$ and *specification* $g$ as $\mathrm{sys}_\varphi$ and $\mathrm{spec}_\varphi$, and similarly the *domain* $X$ and *codomain* $Y$ as $\mathrm{dom}(\varphi)$ and $\mathrm{cod}(\varphi)$. A *compound task* $\Phi$ (or just *task*) is a non-empty set of tasks. As we have seen in previous examples, we notate a task $\Phi$ as a collection of atomic tasks $\mathrm{sys}_\varphi \leftrightharpoons \mathrm{spec}_\varphi$, where superscripts on the harpoons indicate which learnable parameters are governed by each atomic task.

Tasks become concrete objective functions by a hyperparametric choice of divergences for atomic tasks, followed by a combination via a weighted sum with hyperparameter coefficients, or more generally a *compound function*. As such, a particular objective function that instantiates a task is one where the choices for the measure of statistical divergence and compound function have been made. Beyond these hyperparameters, tasks and objective functions may be viewed as informationally equivalent.

**Definition 2.2** (Objective function). Let $\Phi = \{(f_i, g_i, \mathcal{X}_i, \mathfrak{p}_i)\}_{i \in N}$ be a compound task with $N$ atomic tasks. Let $l \in \Sigma$ be a learner of $\Phi$. An *objective function* for $l$ is a tuple $(\Phi_l, \mathcal{D}, \alpha)$ where $\Phi_l = \{(f_i, g_i, \mathcal{X}_i, \mathfrak{p}_i) \mid \text{para}(l) \subseteq \mathfrak{p}_i\}$ is the set of all tasks on which $l$ is optimised, $\mathcal{D}$ is a set of *statistical divergences* $\mathbf{D}_{(\varphi \in \Phi_l)} : \text{cod}(\varphi) \times \text{cod}(\varphi) \to \mathbb{R}^{\geq 0}$, and the *compound function* $\alpha$ is a function $(\mathbb{R}^{\geq 0})^{\times |\Phi_l|} \to \mathbb{R}^{\geq 0}$ that is differentiable, and typically non-decreasing in each argument.

## 2.2 Patterns are "nice" tasks

We do not expect there to be a general and systematic method to translate between natural-language behavioural specifications and tasks; if such a method existed, then all of deep learning would be reduced to hyperparameter search. There are often many different ways to approach a problem in ML, much like there is no single correct way to write software, or design a building. This suggests to us the view of *patterns*: some tasks are well understood, usable modularly and in many contexts, and easily modifiable, and such tasks can be viewed as *design patterns* – borrowing a term from software engineering (Beck & Cunningham, 1987)[1]. In this section, we suggest some examples of patterns that correspond to well-studied methods and paradigms in ML, and we show how to use **patterns as an accessible basis to analyse models.**

**Pattern 2.3** (classification).

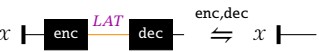

Given a data-label pair $(d, l)$ drawn from $\mathcal{X}$ with labels $l \in L$, a *classifier* cls : $D \to L$ is a function that solves the classification task, in which it seeks to reconstruct the label from the data. This can be done by minimising the corresponding objective function $\mathbb{E}_{(d,l) \sim \mathcal{X}}[\mathbf{D}(\text{cls}(d), l)]$ for some measure of statistical divergence $\mathbf{D}$ on the label space, which may be continuous to encompass regression.

**Pattern 2.4** (autoencoding).

As we have seen, given a data distribution $\mathcal{X}$ over $X$ and some latent space $LAT$, an *autoencoder* consists of an *encoder* enc : $X \to LAT$ and a *decoder* dec : $LAT \to X$ which cooperate to reconstruct the identity over the observed distribution, by minimising $\mathbb{E}_{x \sim \mathcal{X}}[\mathbf{D}(\text{dec}(\text{enc}(x)), x)]$.

**Pattern 2.5** (GAN). Given a data distribution $\mathcal{X}$ over $X$ and noise distribution $\mathcal{N}$ over $N$, a *generative adversarial network* (GAN) consists of a *generator* gen : $N \to X$ and a *discriminator* dsc : $X \to [0, 1]$. The prosaic explanation that "the discriminator seeks to distinguish real data from fake data while the generator aims to fool the discriminator" translates directly into a task description: where 1 indicates "real" and 0 indicates "fake", the discriminator dsc seeks to minimise some positive combination of the terms $\mathbb{E}_{x \sim \mathcal{X}}[\mathbf{D}(\text{dsc}(x), 1)]$ and $\mathbb{E}_{x \sim \mathcal{X}}[\mathbf{D}(\text{dsc}(\text{gen}(x)), 0)]$, while the generator gen seeks to minimise $\mathbb{E}_{x \sim \mathcal{X}}[\mathbf{D}(\text{dsc}(\text{gen}(x)), 1)]$.

---

[1]And before that, architecture and urban design (Alexander et al., 1977).

## 2.3 ANALYSING COMPLEX TASKS

**We can analyse the intended functions of models by viewing them as composites of simple patterns.** We have already seen in Example 1.4 how a VAE is analysable by inspection as a specialised regular autoencoder. As a second example, a `CycleGAN` is two `GAN`s on different distributions, whose generators are mutually autoencoders by a *cycle consistency loss* (Zhu et al., 2017). This suggests that the generators encode the distributions into each other in a reversible manner, and indeed this kind of style transfer between distributions is what a `CycleGAN` does in practice.

**Example 2.6** (`cycleGAN`)**.** Below, $i, j$ are nonequal indices taking values in $\{0, 1\}$, where $\mathcal{X}_0$ and $\mathcal{X}_1$ are different distributions on the same space $X$: typically these are two classes of images.

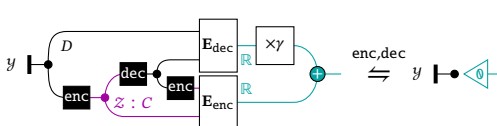

**Tasks allow us to reason "legalistically" to rule out undesirable model behaviours.** For instance, in Example 1.4, we can immediately determine that the normalisation term imposes a nontrivial constraint: without the normalisation task, `enc` and `dec` could collude against us by only using variance 0 (i.e. deterministic) representations that are far apart, which would lose the ease of sampling and robustness of representations. We further elaborate on the use of this form of reasoning for informing task design in Appendix B.

**We can also upgrade informal intuitions into formal derivations.** For example, on the account of Ranzato et al. (2007), a broad class of unsupervised learning techniques — including PCA and $k$-means — are specialisations of the `energy minimisation` task, which may be considered an `autoencoding` task from a latent "code" space subject to the representations minimising a measure of "energy". We can in fact formalise the relationship between these forms of unsupervised learning and `autoencoding`, showing that under mild assumptions, they are the same in the computational limit.

**Pattern 2.7** (`energy minimisation`)**.** `energy minimisation` consists of; three types of systems: $D$(ata), $C$(ode), and $\mathbb{R}^{\geq 0}$; two learnable processes: an encoder $\mathsf{enc} : D \to C$ and a decoder $\mathsf{dec} : C \to D$; two user-supplied *energy functions*: $\mathbf{E}_{\mathsf{enc}} : C \times C \to \mathbb{R}^{\geq 0}$ and $\mathbf{E}_{\mathsf{dec}} : D \times D \to \mathbb{R}^{\geq 0}$; and a user-supplied constant $\gamma \in \mathbb{R}^{\geq 0}$. Provided a distribution of inputs $\mathcal{Y}$ on $D$, and a distribution $\mathcal{Z}$ on $C$ the system seeks to minimise $\mathbb{E}_{y \sim \mathcal{Y}, z \sim \mathcal{Z}}[\gamma \mathbf{E}_{\mathsf{enc}}(\mathsf{enc}(y), z) + \mathbf{E}_{\mathsf{dec}}(y, d(z))]$. Such a task is called *code-extracting* when $\mathcal{Z} = \mathsf{enc}(\mathcal{Y})$.

To formally relate code-extracting `energy minimisation` and `autoencoding`, we introduce a relationship between tasks called *refinement*, which states that perfectly solving $\Phi$ allows one to construct perfect solutions for $\Psi$. When $\Phi$ and $\Psi$ refine each other, the tasks are "the same in the computational limit"; a perfect autoencoder is a perfect code-extracting energy minimiser, and vice versa. These relationships are a proxy for the behaviour and relative power of concrete implementations of tasks.

**Definition 2.8** (Refinement and equivalence of tasks)**.** Task $\Phi$ *refines* task $\Psi$ if, by treating the atomic tasks as equations, the processes of $\Phi$ may be composed to satisfy the equations of $\Psi$. $\Phi$ and $\Psi$ are *equivalent* if they refine one another, which we denote $\Phi \equiv \Psi$.

**Proposition 2.9.** If `enc` and `dec` are deterministic and $\mathbf{E}_{\mathsf{enc}}$ and $\mathbf{E}_{\mathsf{dec}}$ are positive (e.g. metrics or statistical divergences), then `energy minimisation` $\equiv$ `autoencoding`. (proof in Section C.1)

## 3 EXPERIMENT: THE `MANIPULATION` TASK

In this section, we design and experimentally validate `manipulation`, **a novel task which aims to view and edit a property of data without explicit guidance**. There are many models that behave like manipulators, e.g. unsupervised sentiment translation (Li et al., 2018; Sudhakar et al., 2019) in the text domain and prompt-based photo editing (Hertz et al., 2022; Kawar et al., 2023) in the image domain. To constrain the behaviour of such a manipulation, we demonstrate another ability of our framework: importing insights from computer science more broadly via the process-theoretic perspective. For `manipulation`, we reference the field of Bidirectional Transformations, which studies consistency between different overlapping representations of data (Abou-Saleh et al., 2018). A special case is the view-update problem (Bancilhon & Spyratos, 1981) originally proposed for databases: how do we algebraically characterise reading-out and updating an attribute $a \in A$ from some data $d \in D$? There are a family of solutions called *lenses* which are parameterised by algebraic laws of varying strength (Nakano, 2021), which we take inspiration from below:

**Task 3.1** (`manipulation`). Let $\mathcal{X} : (d, a)$ be a distribution over some data $d \in D$, each labelled with an attribute $a \in A$ and let $\mathcal{A}$ be a distribution over the attributes. A `manipulation` consists of a pair of operations ($\text{get} : D \to A$, $\text{put} : D \times A \to D$) which can be understood as reading and writing, respectively. In particular, the `put` edits a reference data point to exhibit the specified attribute. The two operations have to obey the following tasks, with respect to the modeller's choice of distribution $\mathcal{A}$ on $A$.

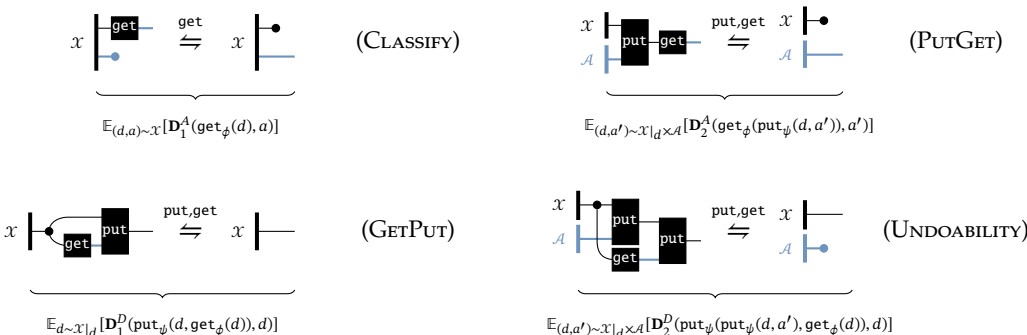

To provide an intuition for each of the tasks, assume that the data consists of images each containing a single shape each labeled with the colour of the shape. CLASSIFY allows us to use `get` to read out the colour of a shape. PutGet says that first editing the colour of a shape (say, from red to blue) and then immediately reading out that colour will return the edited colour (blue). GETPUT says that reading out the colour of a shape (say, red) followed by editing the shape to have the same colour (i.e., an edit that leaves red unchanged) is the same as doing nothing. UNDOABILITY says that edits can be undone; using the first `put` to change the colour of a shape (say from red to blue), and then editing again with a second `put` to restore the original, read-out colour of the shape (red) must restore the original image.

For completeness, and to illustrate the ergonomic necessity of our diagrammatic notation, we display the formulaically obtained hyperparameterised objective function of `manipulation` in standard notation below:

$$\text{argmin}_{\phi,\psi} \left( \begin{array}{c} \alpha \mathbb{E}_{(d,a)\sim\mathcal{X}} \left( \mathbf{D}_1^A \left( \text{get}_\phi(d), a \right) \right) + \beta \mathbb{E}_{(d,a')\sim\mathcal{X}|_d\times\mathcal{A}} \left( \mathbf{D}_2^A \left( \text{get}_\phi \left( \text{put}_\psi(d, a') \right), a' \right) \right) \\ \underbrace{\qquad\qquad\qquad\qquad}_{\text{CLASSIFY}} \qquad\qquad \underbrace{\qquad\qquad\qquad\qquad}_{\text{PutGet}} \\ + \gamma \mathbb{E}_{d\sim\mathcal{X}|_d} \left( \mathbf{D}_1^D \left( \text{put}_\psi \left( d, \text{get}_\phi(d) \right), a \right) \right) + \delta \mathbb{E}_{(d,a')\sim\mathcal{X}|_d\times\mathcal{A}} \left( \mathbf{D}_2^D \left( \text{put}_\psi \left( \text{put}_\psi(d, a'), \text{get}_\phi(d) \right) \right) \right) \\ \underbrace{\qquad\qquad\qquad}_{\text{GETPUT}} \qquad\qquad\qquad \underbrace{\qquad\qquad\qquad\qquad\qquad}_{\text{UNDOABILITY}} \end{array} \right)$$

Where; $\phi, \psi$ are the parameters of `put` and `get` to be learnt; $\alpha, \beta, \gamma, \delta$ are hyperparametric positive weighting coefficients for each task; $\mathbf{D}_1^A, \mathbf{D}_2^A$ are hyperparametric statistical divergences

for $A$; $\mathbf{D}_1^D, \mathbf{D}_2^D$ are statistical divergences for $D$; and $\mathcal{A}$ is a hyperparametric distribution over $A$ such that supp($\mathcal{A}$) contains the attributes the modeller intends to have as targets.

The following pair of theoretical results allow us to anticipate and justify the efficacy of `manipulation`, by analysing its behaviour with respect to an additional regularisation term we detail in Appendix B; in the same way that a VAE may be viewed as a nicely regularised autoencoder, we may learn some things indirectly about `manipulation` by considering a nicely-regularised variant. First, we may relate the behaviour of `manipulation` to style-transfer.

**Proposition 3.2** (`manipulation` vs. style-transfer (Take 1))**.** There exists a regularisation term for `manipulation` such that an "optimally-trained" `manipulation` yields an "optimally-trained" `CycleGAN`. (Elaboration of conditions and proof in Section C.2)

Second, we have a result that informs us that in the computational limit, `manipulation` allows us to transform any classifier of attributes into a "minimally invasive" editor of those attributes, without providing any additional information in the form of augmented data or inductive biases.

**Proposition 3.3** (`put` as Bayesian inverse of a classifier)**.** There exists a regularisation term for `manipulation` such that by identifying a given "well-behaved" classifier `cls` as `get`, satisfaction of the `manipulation` task allows `put` to induce the Bayesian inversion `cls`$^\dagger$. (Elaboration of conditions and proof in Section C.3)

Indeed, as we proceed to report, we obtain such minimal-editors in practice. Moreover, we do so in a manner that is deterministic (cf. VAEs, Kingma & Welling (2022)), non-adversarial (cf. GANs, Goodfellow et al. (2020)), and without hardcoding or handcrafting features in latent spaces (cf. CVAEs, Shaikh et al. (2022)).

### 3.1 Experimental Results I: Simple attributes of synthetic and real-world data

**N.B.** For space, only results are reported in the main body, while all methodological details are reported in Appendix D; as we follow our recipe, these details consist chiefly of hyperparameter choices.

We demonstrate initial proofs-of-concept of the `manipulation` task on a simple analytic dataset (Figure 1) inspired by Spriteworld (Watters et al., 2019), and on MNIST (Figure 2). In the former, each image depicts a single shape with varying properties, and is labelled by two attributes: shape – circle, square or triangle – and colour – red, green or blue. For each attribute, we train a `get`/`put` pair according to the `manipulation` task specification.

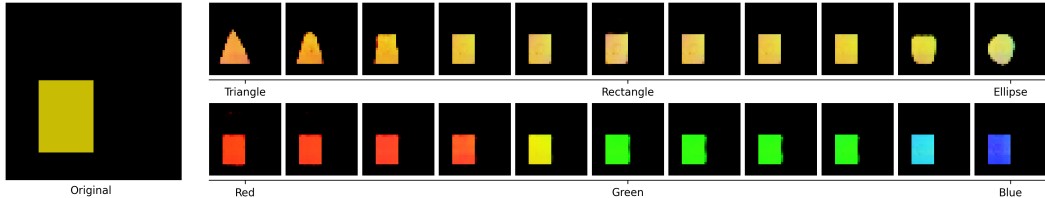

Figure 1: An input Spriteworld image alongside a spectrum of outputs exhibiting the ability of the `put` to manipulate a single attribute of the input while preserving its other properties. Additionally, the model is able to generalise by interpolating to attribute values unseen during training, in this case producing orange and cyan shapes, whereas during training, it only sees red, green or blue shapes. (further details in Section D.1)

### 3.2 Experimental Results II: Derived attributes of synthetic data

Often in practice we are interested in complex, non-explicit attributes that are derived from those labelled in the data: for example, "eligibility for a loan" may be derived from other explicit attributes of people in a database by an operationally opaque classifier, with

Figure 2: Outputs of a `put` trained against an MNIST classifier. The `put` preserves several graphological aspects, such as stroke weight, slant, and angularity. This represents qualitative evidence to support our prediction that `put` as a class-conditioned generative model behaves as a style-preserving edit.

unknown range, distribution, and dependencies on other attributes. A known challenge in manipulating derived attributes is unequal entropy in attribute classes (Chu et al., 2017), which may cause models such as CycleGANs to hide data imperceptibly, making them particularly vulnerable to adversarial attacks. Various solutions have been proposed, including masks (Wu et al., 2024), blurring (Fu et al., 2019) and compression (Dziugaite et al., 2016). We demonstrate via a modification of `manipulation` (Task 3.1,Figure 4) that **our framework permits the design and implementation of end-to-end approaches to editing complex attributes without interventions on the data.**

$$bc = \begin{cases} \min(1, cs + 0.6) & \text{if } shape = circle \\ \min(0.8, \max(0.2, cs)) & \text{if } shape = square \\ \max(0, cs - 0.6) & \text{if } shape = triangle \end{cases}$$

Figure 3: To illustrate the concepts of derived attributes and unequal entropy, consider an attribute on the Spriteworld data called *blue-circleness*, which broadly measures how similar a shape is to a blue circle. We define *blue-circleness* (*bc*) as a function of explicit attributes *shape* and *colour*; we assign a continuous colour score $cs \in [0, 1]$ based on the hue, where red = 0 and blue = 1. To illustrate unequal entropy in this example, the class 0 has higher entropy than 0.4 because there are more shapes that have *bc*-value 0. So manipulating a shape with *bc*-value 0 to 0.4 must lose information.

**Task 3.4** (`complement manipulation`).

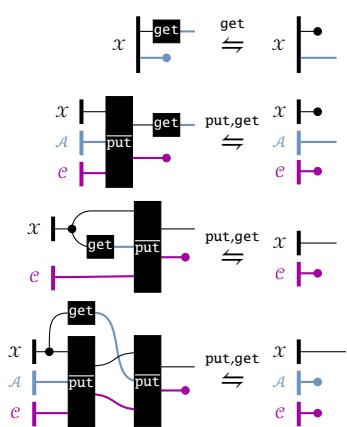

Inspired by the complement of symmetric lenses (Hofmann et al., 2011), we introduce a complement $C$ to put, changing its type to $S \times L \times C \to S \times C$. Let $(d, a) \sim \mathcal{X}$ be a distribution over some data $d \in D$, each labelled with an attribute $a \in A$. `complement manipulation` consists of a pair (`get` : $D \to A$, $\overline{\text{put}}$ : $D \times A \times C \to \mathcal{D} \times C$) fulfilling the rules on the left. The idea of the complement is that it provides the manipulation with a scratchpad $C$ to keep track of additional data. As none of the tasks check the output of the complement, the put and get can use it freely to store relevant data.

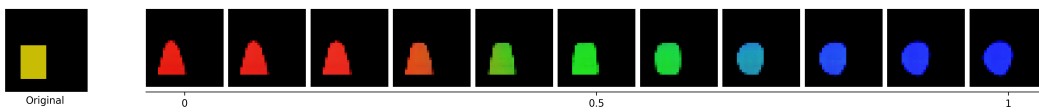

Figure 4: Complement manipulators (Task 3.4) can manipulate derived attributes such as *blue-circleness*, by using the complement as a scratchpad to record a correspondence between data points (further details in Section D.1) while preserving attributes such as position and size.

### 3.3 EXPERIMENTAL RESULTS III: INTERPRETABILITY APPLICATIONS ON REAL-WORLD DATA

As a further test of the robustness of tasks to implementation choices, we specialised the put to be a simple vector addition in the latent space of an autoencoder (Figure 5), for the relatively complex *Large-scale CelebFaces Attributes* dataset. We found that restricting put to be linear in this way increased training stability. Moreover, in the same way that we would expect a latent space "filtered through" the probabilistic structure of Gaussians to yield good sampling properties (Example 1.4), we would predict that enforcing linear structure on the latent space would yield "linear" properties. Indeed, we exhibit continuous interpolation in generated outputs between normally discrete class labels (Figure 6), and class-sensitive separation of latent space embeddings in the autoencoder. We consider this to be compelling evidence that **since our framework is agnostic, implementation details may be engineered to obtain additional desirable properties without compromising behavioural specifications.**

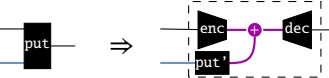

Figure 5: Recalling that architectural choices are a form of specialisation by diagrammatic substitution, the linear put is a specialisation of a generic put as an `autoencoder` task-bound pair of learners `enc` and `dec`, along with a `put'` that computes single shift vector to be added into the latent space, depending only on the label value. `enc`, `dec`, and `put'` are trained simultaneously along with the `manipulation` tasks, and intuitively this pressures the autoencoder pair to adapt their latent representations to fit the needs of the broader `manipulation` task.

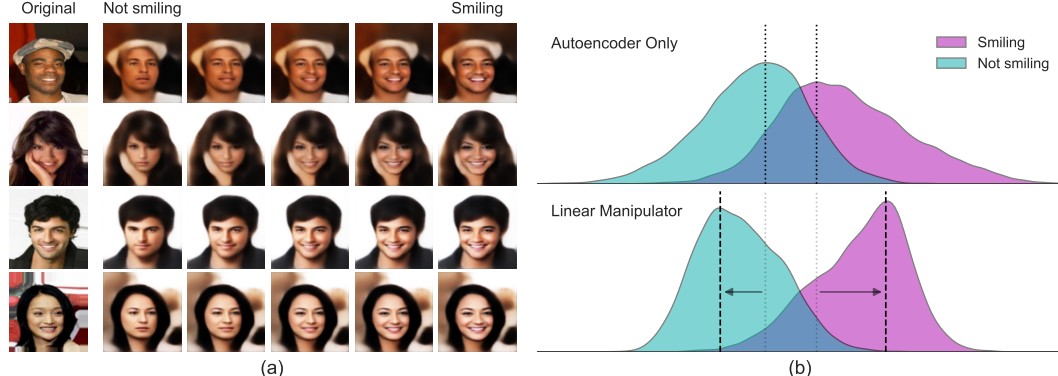

Figure 6: *Left:* Outputs of a linear manipulator trained on face image data paired with a binary "smile"/"no-smile" label. The remarkable aspect of this experiment is that the original data only carried binary smile/no-smile labels, and that the linear structure in the specialisation of the put admits continuous interpolation. *Right:* A comparison of the spread of the latent embeddings of images from the validation set when pre-training an autoencoder and then training a (linear) classifier on the latent space (top), vs. when trained with a linear manipulator (bottom). We find that linear put automatically separates latent space embeddings of classes : the graphs depict the relative density of embeddings along the direction of the classifiers' weight vectors, normalised so that each combined data spread is centred and has unit variance (details in Section D.2).

## 4 CONCLUDING DISCUSSION

### 4.1 SUMMARY

**We introduced a diagrammatic language for representing and reasoning about the behaviour of machine learning models** in terms of tasks, viewed as the essential data of objective functions. By leveraging category theory and string diagrams, our work establishes a cross-disciplinary formal bridge between theoretical computer science and practical machine learning, providing new conceptual tools for analyzing ML systems and permitting the transfer of insights between traditionally separate fields.

The proposed framework allows capturing existing tasks in machine learning, providing intuitive insights rooted in mathematical rigour. We identify a set of widespread and well-understood tasks, which we call patterns. We can analyse some tasks as composites of patterns (Example 2.6) while other tasks can be understood as specialisations of patterns (Example 1.4). The rewrite system inherent to string diagrams, allows us to identify relationships between different tasks and formalise intuitions (Proposition 2.9, Proposition 3.2).

**Beyond theoretical insights, the proposed language also allows the creation of new training paradigms.** As preliminary empirical validation of our theory's utility and potential, we introduced a novel task type called manipulator that produces a class-conditioned and style-preserving generative model counterpart for a given classifier. In the image domain, we were able to verify predicted behaviours (Section 3.1), and we demonstrated the ability to design novel end-to-end capabilities, such as end-to-end editing of complex attributes (Section 3.2) and the imposition of linear structure on latent space representations (Section 3.3), which allowed continuous interpolation between discrete class labels on real data, and separated latent space embeddings of different classes. Notably, this was achieved without adversarial training conditions, random sampling, preprocessing of data, or hardcoded interventions in the architectures, i.e.:

**Our framework enables capable, small-scale, and training-stable models.**

## 4.2 RELATION TO EXTANT WORK

Regarding our nascent theoretical framework as a whole, the style of engineering beginning from tasks is already common practice in many fields of ML, and we sought here to place these practices on a more rigorous footing, and to probe their strengths and limitations. Our mathematical approach draws broadly from the field of Applied Category Theory (Fong & Spivak, 2019a), particularly in the use of string diagrams for the higher-algebraic data of concurrently and serially composed functions, which enables compact representation and reasoning with otherwise cumbersome symbolic equivalents when dealing with multiple learners in tandem. To our knowledge, our concern with the composition of tasks among many learners distinguishes our aims and formal choices from approaches that employ similar mathematical formulations, both within the category-theoretic literature (cf. Gavranović et al. (2024)), and without (cf. the variational generalisation of Bayesian inference presented in Knoblauch et al. (2022)).

While our approach is essentially neurosymbolic in spirit, it does not fit neatly into the mainstream triad of neurosymbolic approaches (d'Avila Garcez & Lamb, 2020); we do not encode symbolic data for neural operations, nor do we interface neural approaches with symbolic engines, nor are we hardcoding expert knowledge representations. Moreover, our aims differ: while neurosymbolic approaches often seek to manipulate symbolic data systematically by neural means, our framework operates at a higher level of abstraction, seeking to use the systematicity of higher-algebraic equational characterisations as a means to shape the neural ends. Hence our perspective may complement existing approaches to structure in machine learning.

Regarding `manipulation` in particular, this was to our knowledge the first practically demonstrated synthesis of insights from Bidirectional Transformations as a subfield of database theory (Abou-Saleh et al., 2018) with ML. While explicitly neurosymbolic approaches have been tried for similar editing tasks before (see e.g. Smet et al. (2023)), owing to the influence of database theory in our approach, to our knowledge our statement and execution of this task enjoys the maximal permissible generality and implementation agnosticism among similar attribute-editing tasks, without sacrificing rigour and systematicity.

## 4.3 LIMITATIONS AND PROSPECTS

Concerning `manipulation` in particular, an immediately evident limitation of this practical demonstration is a lack of exploration of how the difficulty of training such ensembles of learners behaves at scale, with respect to more complex and multimodal datasets, and with a wider range of architectures. Concerning scale, while none of the products of our experimentation are state-of-the-art with respect to specific applications, we believe the variety and promise of these results serve as a compelling validation of our theoretical framework's utility and potential. Concerning applications of `manipulator` beyond the image domain, we report on some sketch experiments in sentiment-manipulation on text in Appendix F, where we also comment on the nature of technical difficulties to be overcome in the application of our framework to complex domain data, and offer an explanation for mode collapses observed during training by empirically relating `manipulator` to other generative classification approaches. Concerning a wider range of architectures, we report on some specialisations of the learners. However, we leave exploring `manipulation` in combination with state-of-the-art architectures, such as diffusion, for future work.

Addressing the theoretical framework of tasks more broadly, our reliance on equational characterisations is double-edged. On one hand, it is uncommon to find such characterisations of mathematical systems of interest as they are usually defined by more direct means, and this presents a theoretical limitation. On the other hand, it appears that the strength of equational characterisations when applied to ML lies in imposing structure on "the way to learn to solve a problem" rather than on the solutions or problems themselves (Sutton, 2019). This suggests promising future possibilities of our mathematical framework in bridging

structural-symbolic approaches from computer science more broadly with methods that can effectively leverage computation; we suggest "neural data structures" as a sketch experiment in Appendix E.

While we have demonstrated that, in certain cases, tasks can determine behaviours, there is a theoretical gap in the converse analysis of behaviours of trained models in terms of their basic tasks. The problem is typified by generative models where it is impossible a priori for all of the constituent tasks to be simultaneously perfectly optimised. The prototypical illustrating example is the GAN, where it is impossible for both the generator and discriminator to be perfect. Conceptually, the gap is that we have only dealt here with "static" ensembles which in principle admit idealised loss-minimisations, whereas some generative models use adversarial tasks to enforce "dynamic" training forces for a variety of purposes, such as representation regularisation for easier sampling in VAEs. While we do offer some initial methods of analysis in Section C.2, a more thorough and encompassing analysis is beyond the scope of this paper.

Future theoretical developments will seek to incorporate other aspects of ML: for example, relating to work that focuses on the choice of model architecture (Khatri et al., 2024) and interactions with the underlying data distribution (Bronstein et al., 2021). While our current experiments focus on demonstrating our framework's validity, future practical developments will explore applications to more complex, real-world ML challenges, where we envision our approach informing areas such as AutoML, interpretable AI, and formal verification of ML systems: the compositional nature of our task-based framework naturally aligns with neural architecture search by potentially informing principled search strategies for optimal model architectures, and the explicit representation of model behaviours as equational constraints could enhance interpretability and facilitate formal verification.

Ethics Statement

We recognise the potential impact of ML, both positive and negative, on global *society* and *nature* as ethical stakeholders.

Regarding society, we recognise that expertise in, and steering of, the development and implementation of ML systems is concentrated in the hands of relatively few. Hence, we believe that broad democratisation of methods and understanding would be an improvement upon current circumstances. Accordingly, in the development of our theory and exposition, we were biased away from formal rigidity in favour of accessibility and expressive possibility. Our "pattern language" approach seeks to be a minimally constraining mode of communication, reasoning, and design.

Regarding nature, we recognise that practical ML carries a growing ecological footprint, primarily due to the energy-intensive nature of training large models and running extensive experiments. We believe that a compositional approach could enable efficient and smaller model design, reducing environmental impact.

By formalising ML tasks in a way that allows for more principled and targeted experimentation, we aim to encourage a more thoughtful, resource-conscious, and participatory approach to ML research and development. We acknowledge that there is still much work to be done in this regard, and we hope that our framework will inspire further research into sustainable and harm-free ML practices.

Reproducibility Statement

Experiment methodology and details such as model parameters are reported in the Appendix. All code is available in the supplementary materials, and will be made public.

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

## A    String diagrams for tasks

String diagrams are a formal diagrammatic syntax that take semantics in symmetric monoidal categories, and they find usage in a broad variety of fields[2]. Our string diagrams are built using sequential and parallel composition from the following generators, along with a stock of function labels:

| | | | | |
|---|---|---|---|---|
| $\mathbb{R}^M - \boxed{f} - \mathbb{R}^N$ | $\{\star\}$ | $\mathbb{R}^M - \bullet \cdots \{\star\}$ | $\cdots x \vdash \mathbb{R}^M$ | $\mathbb{R}^A - \boxed{f} - \boxed{g} - \mathbb{R}^C$ |
| Function | Singleton space | Delete | Data from $x$ | Sequential composition |
| $f : \mathbb{R}^M \to \mathbb{R}^N$ | $\mathbb{R}^0 \simeq \{\star\}$ | $\epsilon : \mathbb{R}^M \to \{\star\}$ | $x : \{\star\} \to \mathbb{R}^M$ | $(g \circ f) : \mathbb{R}^A \to \mathbb{R}^C$ |

| | | | | |
|---|---|---|---|---|
| $\{\star\} \cdots \triangleleft - \mathbb{R}^N$ | Swap | $\mathbb{R}^M - \bullet$ | $\cdots x \vdash \mathbb{R}^M$, $\cdots y \vdash \mathbb{R}^N$ | Parallel composition |
| Vector | Swap | Copy | Independent $x, y$ | Parallel composition |
| $v \in \mathbb{R}^N$ | $\theta : \mathbb{R}^{(N+M)} \to \mathbb{R}^{(M+N)}$ | $\Delta : \mathbb{R}^M \to \mathbb{R}^{(M+M)}$ | $(x \times y) : \{\star\} \to \mathbb{R}^{(M+N)}$ | $(f \oplus g) : \mathbb{R}^{(A+C)} \to \mathbb{R}^{(B+D)}$ |

For conventional reasons that were not by our choice, vectors are depicted as triangular nodes with only output wires, reminiscent of bra-ket notation. (co)associative (co)monoids, such as copy-delete and add-zero, are specially depicted as circular nodes as is common in applied category theory. In this work, encoders and decoders are sometimes depicted as "bottlenecking" trapezia, as is common in ML, and distributional states are given their own notation as thick bars.

An attractive characteristic of string diagrams is that visually intuitive equivalences between information flows are guaranteed to correspond to symbolic derivations of behavioural equivalence: tedious algebraic proofs of equality between sequentially- and parallel-composite processes are suppressed and absorbed by (processive) isotopies of diagrams. In the diagrammatic syntax it is conventional to notate such isomorphisms as plain equalities. Interested readers are referred to (Selinger, 2011) for the relevant mathematical foundations.

$$(\mathbf{1} \oplus \theta) \circ (\Delta \oplus g) \circ (f \oplus \mathbf{1})$$
$$\simeq (\mathbf{1} \oplus \theta) \circ (\mathbf{1} \oplus \mathbf{1} \oplus g) \circ (\Delta \oplus \mathbf{1}) \circ (f \oplus \mathbf{1}) \quad \text{[Identity, interchange]}$$
$$\simeq (\mathbf{1} \oplus g \oplus \mathbf{1}) \circ (\mathbf{1} \oplus \theta) \circ (\Delta \oplus \mathbf{1}) \circ (f \oplus \mathbf{1}) \quad \text{[Braid naturality]}$$
$$\simeq (\mathbf{1} \oplus g \oplus \mathbf{1}) \circ (\mathbf{1} \oplus \theta) \circ (f \oplus f \oplus \mathbf{1}) \circ (\Delta \oplus \mathbf{1}) \quad \text{[Copy naturality]}$$
$$\simeq (\mathbf{1} \oplus g \oplus \mathbf{1}) \circ (f \oplus \mathbf{1} \oplus f) \circ (\mathbf{1} \oplus \theta) \circ (\Delta \oplus \mathbf{1}) \quad \text{[Braid naturality]}$$

### A.1    Categorical semantics of task diagrams

The functional effect of the construction below is to extend the category of continuous maps between Euclidean spaces with global elements that behave as probability distributions instead of points. We presume familiarity with symmetric monoidal categories and their graphical calculi (Selinger, 2011).

Let **CartSp** denote the coloured PROP (Yau, 2008) of continuous maps between Euclidean spaces, where the tensor product is the cartesian product — i.e. **CartSp** is cartesian monoidal.

Let **BorelStoch** denote the Markov category (Cho & Jacobs, 2019; Fritz, 2020) of stochastic kernels (Panangaden, 1999) between Borel-measurable spaces. Stochastic kernels in particular subsume the continuous maps between Euclidean spaces.

As a Markov category, in the terminology of (Fong & Spivak, 2019b), **BorelStoch** supplies cocommutative comonoids. By Fox's theorem (Fox, 1976b) cartesian monoidal categories

---

[2]Including linear and affine algebra (Sobociński, 2015; Bonchi et al., 2017; 2019), first order logic (Haydon & Sobociński, 2020), causal models (Lorenz & Tull, 2023; Jacobs et al., 2019), signal flow graphs (Bonchi et al., 2014), electrical circuits (Boisseau & Sobociński, 2022), game theory (Hedges, 2015), petri nets (Baez & Master, 2020), probability theory (Fritz et al., 2021), formal linguistics (Coecke et al., 2010; Wang-Mascianica et al., 2023; Wang-Maścianica, 2023), quantum theory (Coecke & Duncan, 2011; Coecke & Kissinger, 2017; Poór et al., 2023), and aspects of machine learning such as backpropagation (Cruttwell et al., 2022).

are precisely those isomorphic to their own categories of cocommutative comonoids. Hence there is a (semicartesian) functorial embedding of **CartSp** into **BorelStoch** sending $\mathbb{R}^N$ to $\mathbb{R}^N$ (equipped with the usual Borel measure), and continuous maps to deterministic continuous maps. **We declare our semantics** to be taken in the category to be generated by the image of this embedding along with the probability distributions $\mathcal{X} : \{\star\} \to \mathbb{R}^N$, where $\{\star\}$ is the singleton monoidal unit of **BorelStoch**.

### A.2 CATEGORICAL SEMANTICS FOR IDEALISED UNIVERSAL APPROXIMATORS

As we are concerned with behaviour, not implementation details, we idealise all neural networks as perfect universal approximators, which we may formulate string-diagrammatically in a monoidal closed category, borrowing evaluator-notation from (Pavlovic, 2013; 2023). In essence, we are assuming that architectures are sufficiently expressive to optimise whatever tasks we give them; in practice, the conditions under which architectures become universal approximators can be mild (Hornik et al., 1989), and the idealisation is increasingly true-in-practice in the contemporary context of increasing data and compute.

**Definition A.1** (Learner). Let $X, Y$ denote input and output types. A process $\Omega : \mathfrak{p} \oplus X \to Y$ with parameters in $\text{para}(\Omega) = \mathfrak{p} = \mathbb{R}^n$ (for sufficiently large $n$) is a *universal approximator* or *learner* when[3]:

$$\forall f_{X \to Y} \exists \hat{f}_{\in \mathfrak{p}} : \quad \boxed{X\ f\ Y} \quad = \quad \text{[diagram]}$$

The parameter space could represent e.g. the phase space of weights and biases of a neural network.

**Example A.2.** By visual convention, we use colours to indicate different data types of wires. We depict processes with no free parameters as white boxes, and learners as black-boxes with variable labels to indicate distinct or shared parameters. The following composite process has one function $f$ with no learnable parameters, and three neural nets: the two labelled $\alpha$ share a parameter in the space $\mathfrak{p}$, and the one labelled $\beta$ takes a parameter in $\mathbb{Q}$. In this paper, we favour the shorthand on the right.

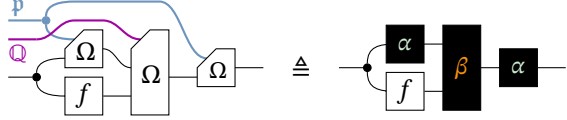

The universal approximation theorem, suitably idealised, manifests as the capacity for a black-box learner to be diagrammatically substituted for any other composite diagram with equal input and output, including those composites that contain other learners. For example, recall the linear `put` below, which may be viewed as substituting a particular composite of `put'`, `enc`, `dec`, and addition in place of `put`:

The universal approximation theorem, suitably idealised, manifests as the capacity for a black-box learner to be diagrammatically substituted for any other composite diagram with equal input and output, including those composites that contain other learners. For example, recall the linear `put` below, which may be viewed as substituting a particular composite of `put'`, `enc`, `dec`, and addition in place of `put`:

This ability is referred to in this work intermittently as *specialisation*, and as *expressive reduction* in (Khatri et al., 2024) where the concept first appeared. For the sake of completeness, we reproduce the relevant construction that gives category-theoretic semantics to universal approximators and specialisation below, along with standard definitions, with the authors' permission.

**Definition A.3** (PROP). A PROP is a strict symmetric monoidal category generated by a single object $x$: every object is of the form

$$\bigotimes^{n} x = x \underbrace{\otimes \cdots \otimes}_{n} x$$

---

[3]We adapt the shape of the universal approximators to clearly indicate the parameters.

PROPs may be generated by, and presented as *signatures* $(\Sigma, E)$ consisting of generating morphisms $\Sigma$ with arity and coarity in $\mathbb{N}$, and equations $E$ relating symmetric monoidal composites of generators.

**Definition A.4** (Coloured PROP). A *multi-sorted* or *coloured* PROP with set of colours $\mathfrak{C}$ has a monoid of objects generated by $\mathfrak{C}$.

**Definition A.5** (Cartesian PROP). By Fox's theorem (Fox, 1976a), a cartesian PROP is one in which every object (wire) is equipped with a cocommutative comonoid (copy) with counit (delete) such that all morphisms in the category are comonoid cohomomorphisms.

**Definition A.6** ((symmetric, unital) coloured operad). Where $(\mathcal{V}, \boxtimes, J)$ is a symmetric monoidal category and $\mathfrak{C}$ denotes a set of *colours* $c_i$, a coloured operad $\mathcal{O}$ consists of:

- For each $n \in \mathbb{N}$ and each $(n+1)$-tuple $(c_1, \cdots, c_n; c)$, an object $\mathcal{O}(c_1, \cdots, c_n; n) \in \mathcal{V}$

- For each $c \in \mathfrak{C}$, a morphism $1_c : J \to \mathcal{O}(c; c)$ called the *identity of $c$*

- For each $(n+1)$-tuple $(c_1 \cdots c_n; c)$ and $n$ other tuples $(d_1^1 \cdots d_{k_1}^1) \cdots (d_1^n \cdots d_{k_n}^n)$ a *composition morphism*

$$\mathcal{O}(c_1, \cdots, c_n; c) \boxtimes \mathcal{O}(d_1^1 \cdots d_{k_1}^1) \boxtimes \cdots \boxtimes \mathcal{O}(d_1^n \cdots d_{k_n}^n) \to \mathcal{O}(d_1^1 \cdots d_{k_1}^1 \cdots d_1^n \cdots d_{k_n}^n; c)$$

- for all $n \in \mathbb{N}$, all tuples of colours, and each permutation $\sigma \in S_n$ the symmmetric group on $n$, a morphism:

$$\sigma^* : \mathcal{O}(c_1 \cdots c_n; c) \to \mathcal{O}(c_{\sigma^*(1)} \cdots c_{\sigma^*(n)}; c)$$

The $\sigma^*$ must represent $S_n$, and composition must satisfy associativity and unitality in a $S_n$-invariant manner.

**Construction A.7** (Hom-Operad of coloured PROP). Where $(\mathcal{P}, \otimes, I)$ is a coloured PROP with colours $\mathfrak{C}_{\mathcal{P}}$, we construct $\mathcal{O}_{\mathcal{P}}$, the *hom-operad* of $\mathcal{P}$. We do so in two stages, by first defining an ambient operad, and then restricting to the operad obtained by a collection of generators. Let the ambient symmetric monoidal category be $(\mathbf{Set}, \times, \{\star\})$. Let the colours $\mathfrak{C}_{\mathcal{O}}$ be the set of all tuples $(\mathbf{A}, \mathbf{B})$, each denoting a pair of tuples $(A_1 \otimes A_n, B_1 \otimes B_n)$ of $A_i, B_i \in \mathfrak{C}_{\mathcal{P}}$.

- The tuple $((\mathbf{A}^1, \mathbf{B}^1) \cdots (\mathbf{A}^n, \mathbf{B}^n); (\mathbf{A}, \mathbf{B}))$ is assigned the set $[\mathcal{P}(\mathbf{A}^1, \mathbf{B}^1) \times \cdots \times \mathcal{P}(\mathbf{A}^n, \mathbf{B}^n) \to \mathcal{P}(\mathbf{A}, \mathbf{B})] \in \mathbf{Set}$; the set of all *generated* functions from the product of homsets $\mathcal{P}(\mathbf{A}^i, \mathbf{B}^i)$ to the homset $\mathcal{P}(\mathbf{A}, \mathbf{B})$.

- $1_{(\mathbf{A}, \mathbf{B})} : \{\star\} \to [\mathcal{P}(\mathbf{A}, \mathbf{B}) \to \mathcal{P}(\mathbf{A}, \mathbf{B})]$ is the identity functional that maps $f : \mathbf{A} \to \mathbf{B}$ in $\mathcal{P}(\mathbf{A}, \mathbf{B})$ to itself.

- The composition operations correspond to function composition in $\mathbf{Set}$, where $[X \to Y] \times [Y \to Z] \to [X \to Z]$ sends $(f_{:X \to Y}, g_{:Y \to Z}) \mapsto (g \circ f)_{:X \to Z}$; appropriately generalised to the multi-argument case. The permutations are similarly defined, inheriting their coherence conditions from the commutativity isomorphisms of the categorical product $\times$.

The generators are:

- For every $f \in \mathcal{P}(\mathbf{A}, \mathbf{B})$ that is a generator of $\mathcal{P}$, define a corresponding generator of type $\{\star\} \to [\mathcal{P}(I, I) \to \mathcal{P}(\mathbf{A}, \mathbf{B})]$, which is the functional $\big(- \mapsto (f \otimes -)\big)$ that sends endomorphisms of the monoidal unit of $\mathcal{P}$ to their tensor with $f$, viewed as an element of the set $[\mathcal{P}(I, I) \to \mathcal{P}(\mathbf{A}, \mathbf{B})]$.

- For every pair of tuples $\big((\mathbf{X}^1, \mathbf{Y}^1) \cdots (\mathbf{X}^m, \mathbf{Y}^m); (\mathbf{A}, \mathbf{B})\big)$ and $\big((\mathbf{J}^1, \mathbf{K}^1) \cdots (\mathbf{J}^n, \mathbf{K}^n); (\mathbf{B}, \mathbf{C})\big)$ in $\mathfrak{C}_{\mathcal{O}}$, a corresponding *sequential composition* operation of type:

$$[\prod_{i \leqslant m} \mathcal{P}(\mathbf{X}^i, \mathbf{Y}^i) \to \mathcal{P}(\mathbf{A}, \mathbf{B})] \times [\prod_{j \leqslant n} \mathcal{P}(\mathbf{J}^j, \mathbf{K}^j) \to \mathcal{P}(\mathbf{B}, \mathbf{C})]$$

$$\to [(\prod_{i\leqslant m} \mathcal{P}(\mathbf{X}^i,\mathbf{Y}^i) \times \prod_{j\leqslant n} \mathcal{P}(\mathbf{J}^j,\mathbf{K}^j)) \to \mathcal{P}(\mathbf{A},\mathbf{C})]$$

Which maps pairs of functionals $(F_{:\prod_{i\leqslant m}\mathcal{P}(\mathbf{X}^i,\mathbf{Y}^i)\to\mathcal{P}(\mathbf{A},\mathbf{B})}, G_{:\prod_{j\leqslant n}\mathcal{P}(\mathbf{J}^j,\mathbf{K}^j)\to\mathcal{P}(\mathbf{B},\mathbf{C})})$ to the functional which sends $p^i : \mathbf{X}^i \to \mathbf{Y}^i$ and $q^j : \mathbf{X}^j \to \mathbf{Y}^j$ to $G(p_1 \cdots p_m) \circ F(q_1 \cdots q_n)$.

- An analogous *parallel composition* for every pair of tuples, which sends pairs of functionals $(F, G)$ to $G(p_1 \cdots p_m) \otimes F(q_1 \cdots q_n)$.

**Remark A.8.** For technical reasons involving scalars (the endomorphisms of the monoidal unit), this construction only works in semicartesian settings, i.e. where the monoidal unit is also terminal, but that is sufficiently general to admit our use cases, which are primarily in cartesian monoidal settings (Fox, 1976a) and semicartesian Markov categories for probabilistic settings (nLab authors, 2024b).

**Example A.9.** Construction A.7 can be thought of as bridging diagrams with their specific algebraic descriptions using just the basic constructors $\circ, \otimes$; the hom-operad (when notated suggestively in the usual tree-notation, found e.g. in Markl et al. (2007)) plays the role of the syntactic tree of $\circ, \otimes$ operators. For instance, given the composite morphism $(g \otimes 1_E) \circ (1_A \otimes f)$ in PROP $\mathcal{P}$, the corresponding diagram and operad-state in $\mathcal{O}_{\mathcal{P}}$ is:

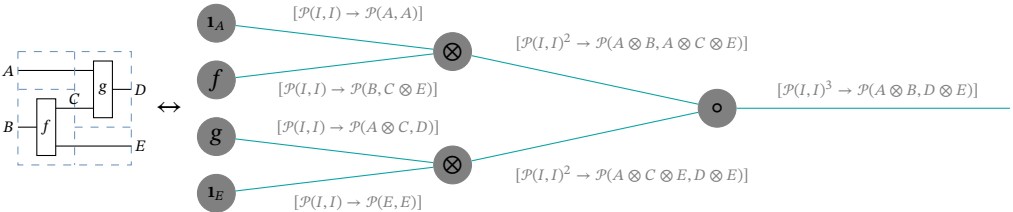

Since the PROPs **CartSp** and its free tensoring are cartesian, $\mathcal{P}(I,I)$ is a singleton containing only the identity of the monoidal unit, so in the settings we are concerned with, we may simplify colours of the form $[\mathcal{P}(I,I)^N \to \mathcal{P}(\mathbf{A},\mathbf{B})]$ to just $\mathcal{P}(\mathbf{A},\mathbf{B})$, and operad-states $\{\star\} \to \mathcal{P}(\mathbf{A},\mathbf{B})$ are in bijective correpondence with morphisms $f : \mathbf{A} \to \mathbf{B}$ of $\mathcal{P}$; the fact that all $f : \mathbf{A} \to \mathbf{B}$ are representable as operad states follows by construction, since any $f$ in $\mathcal{P}$ is by definition expressible in terms of the generators of $\mathcal{P}$, and sequential and parallel composition $\circ, \otimes$. As we assume homsets are already quotiented by the equational theory of $\mathcal{P}$ and the symmetric monoidal coherences, our operadic representations inherit them: for example, we obtain interchange equalities such as the one below for free:

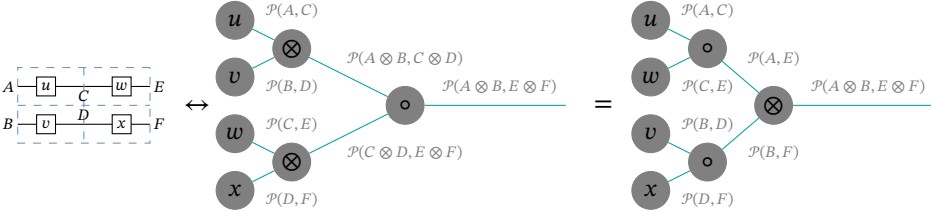

**Definition A.10** (Universal approximators and specialisation). A morphism of a coloured PROP $\mathcal{P}$ of type $(\mathbf{A},\mathbf{B})$ containing universal approximators as black-boxes of types $\mathbf{A}^{i\leqslant n} \to \mathbf{B}^{i\leqslant n}$ is a morphism $((\mathbf{A}^1,\mathbf{B}^1) \cdots (\mathbf{A}^n,\mathbf{B}^n); (\mathbf{A},\mathbf{B}))$ of $\mathcal{O}_{\mathcal{P}}$, and by construction, vice versa. Specialisation corresponds to precomposition in $\mathcal{O}_{\mathcal{P}}$.

**Example A.11.** The inputs of open morphisms in $\mathcal{O}_{\mathcal{P}}$ correspond to "typed holes", and operadic precomposition corresponds to "filling holes", with contents that may themselves also contain typed holes. This precisely formalises the intuition that expressive reductions correspond to the ability of a universal approximator to simulate anything, including

composites containing other universal approximators.

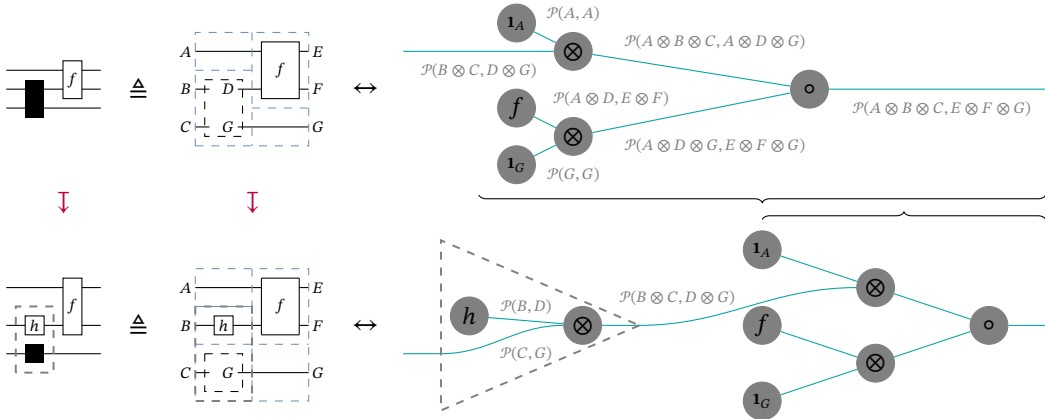

**Remark A.12.** The extension of the current theory to accommodate parameter sharing between universal approximators is conceptually straightforward but technically involved. Parameter sharing corresponds to the ability to reuse – i.e. copy – data between open wires in the operad $\mathcal{O}_{\mathcal{P}}$, which amounts to having a cartesian operad.

## B  STRONG MANIPULATION

The basic `manipulation` admits pathological counterexamples, which we may block by imposing additional tasks. We can treat these new tasks as additional regularisation terms. This section further illustrates a form of "legalistic" thinking using tasks: by thinking of ways that "noncooperative" or "naughty" learners might seek to satisfy tasks without exhibiting the behaviour that the modeller desires. By identifying these counterexamples and constructing additional tasks that block them, the modeller may iteratively improve the behaviour of the model. For illustration, consider the following examples of pathological behaviour that satisfy basic `manipulation`, again in the setting of editing the colour of a shape.

**Example B.1** (Flipping). Consider a `put` that changes the colour of a shape as desired, but then vertically flips the shape. If the classifier `get` is insensitive to the position and orientation of the shape, then CLASSIFY, PUTGET, and GETPUT are satisfied. Moreover, since a vertical flip is its own inverse, composing two `put`s as in the UNDOABILITY task will not detect this aberration. Speaking in more general terms, if there are properties that `get` is insensitive to, there must be additional guardrails to ensure that `put` preserves these other properties as the identity, rather than one of potentially very many self-inverse symmetries.

**Example B.2** (Adversarial decorations). While the classifier `get` may be perfect in-distribution, there are no guarantees about its behaviour out of distribution, for example, when given images with multiple shapes, where it might only classify the leftmost shape. So, it is possible that `put` learns to make edits that take the resulting image out-of-distribution: for example, by adding a red circle next to a blue square to fool the classifier into outputting "red". This would satisfy CLASSIFY, PUTGET, and GETPUT. If the `put` can recognise and undo its own decorations, then UNDOABILITY will also be satisfied. Speaking more generally, we require additional guardrails to ensure that `put` returns something in-distribution.

The `strong manipulation` adds additional tasks to the `manipulator`. A `strong manipulator` has to satisfy the original four tasks plus the following four:

**Task B.3** (`strong manipulation` add-ons)**.**

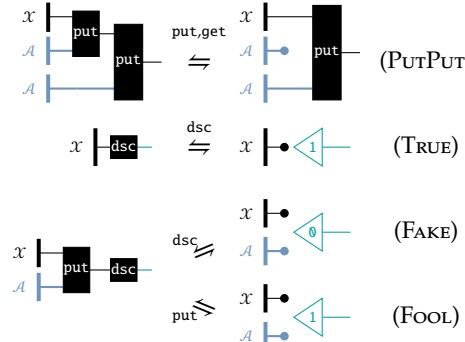

The PUTPUT task (which is strictly stronger than UNDOABILITY in that it is algebraically implied) says that the effect of `put`ting twice is the same as discarding the effect of the first edit and only keeping the last edit. In conjunction with PUTGET and GETPUT, this creates what is known in the literature as a *very well-behaved lens*, which blocks Example B.1 and similar modifications of the data `get` is insensitive to. The TRUE, FAKE, and FOOL tasks introduce a discriminator component `dsc`, which forms a `GAN` pattern with respect to `put` as the generator. When well-trained, this forces the outputs of `put` to lie in-distribution. As in general there are no algebraic or equational laws that characterise arbitrary distributions of data, using `GAN`s in this way is a generic recipe for shaping outputs of generators to behave well in-distribution.

**Remark B.4** (Why basic `manipulation` is preferable in practice)**.** We have observed informally that conditions such as those in basic `manipulation` where learners are cooperative and there are only learners on the LHS appear to be more stable during training. We suggest a sketch reason why: in the tasks of `strong manipulation`, PUTPUT has `put` occur on both the LHS and RHS, which establishes a nontrivial dependence on the current position on parameter-space of `put` in the process of finding a solution. Similarly, the `GAN` rules of `strong manipulation` establishes adversarial mutual dependencies in the parameters of `dsc` and `put`. Conceptually, these dependencies create dynamical systems on the paths that the learners take over the course of training in parameter-space, which may for instance include stable orbits and chaotic behaviour, and may be highly sensitive to initial conditions. A further elaboration of "static" versus "dynamic" tasks in tandem with the ability to express equivalent tasks is potentially useful for creating train-stable models with equivalent behaviour, but this is beyond the scope of this paper, and left for future work.

## C  DEFERRED PROOFS

In this section, we refer to tasks and their realisations interchangeably: so instead of "manipulators" realising the `manipulation` task, we speak just of a `manipulator`, disambiguating when necessary.

### C.1  PROOF OF PROPOSITION 2.9

**Lemma C.1.** For all well-typed $f, g$, and for any positive linear combination $\alpha : \mathbb{R}^{\geq 0} \times \mathbb{R}^{\geq 0} \to \mathbb{R}^{\geq 0}$:

$$x \dashv\!\!\bullet\!\!\left[\begin{matrix} f \\ g \end{matrix}\right]\!\!\alpha\vdash \;\leftrightharpoons\; x \vdash\!\!\bullet\!\!\triangleleft\!\!\boxed{0} \quad \sim \quad x \dashv\!\!\boxed{f}\vdash \;\leftrightharpoons\; x \vdash\!\!\bullet\!\!\triangleleft\!\!\boxed{0} \;\leftrightharpoons\; x \vdash\!\!\boxed{g}\vdash$$

*Proof.* For the forward refinement, as $\alpha$ is a positive linear combination, we have for all $x \in \mathcal{X}$ that $\alpha(f(x), g(x)) = \alpha_1 \cdot f(x) + \alpha_2 \cdot g(x) = 0$. If $f$ and $g$ are constant-functions $0$, we are done. Otherwise, positivity implies that $\alpha_1 \cdot f(x) = \alpha_2 \cdot g(x) = 0$, and since $\alpha_1, \alpha_2 \in \mathbb{R}^{\geq 0}$, then $f(x) = 0 = g(x)$, which is the desired task. For the backwards refinement, if $f(x) = 0 = g(x)$ for all $x \in \mathcal{X}$, then $\alpha(f(x), g(x)) = \alpha_1 \cdot f(x) + \alpha_2 \cdot g(x) = 0$. $\qquad\square$

**Lemma C.2.** For a real-valued pairwise measure $\mathbf{D} : \mathcal{D}(Y) \times \mathcal{D}(Y) \to \mathbb{R}^{\geq 0}$ on the space of distributions over $Y$, the positivity axiom $\mathbf{D}(\mathcal{Y}_1, \mathcal{Y}_2) = 0 \iff \mathcal{Y}_1 = \mathcal{Y}_2$ implies, for (almost[4]) all $f, g : X \to Y$ and $\mathcal{X}$:

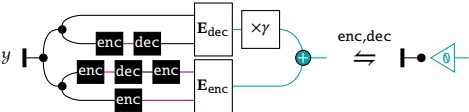

*Proof.* For the forward refinement, we assume that $\mathbf{D}(f(\mathcal{X}), g(\mathcal{X})) = 0$. By positivity of $\mathbf{D}$, $f(\mathcal{X}) = g(\mathcal{X})$, which is the right hand task. For the backward refinement, if $f(\mathcal{X}) = g(\mathcal{X})$, then, by positivity, $\mathbf{D}(f(\mathcal{X}), g(\mathcal{X})) = 0$. $\square$

**Proposition C.3.** If enc and dec are deterministic and $\mathbf{E}_{\text{enc}}$ and $\mathbf{E}_{\text{dec}}$ are positive (e.g. metrics or statistical divergences), then `energy minimisation` $\equiv$ `autoencoding`.

*Proof.* As enc and dec are functions, we may copy them through their outputs to re-express `energy minimisation` as:

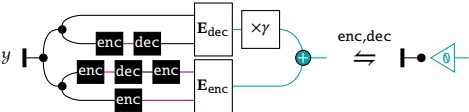

By Lemma C.1, this is equivalent to:

Recall that $\mathbf{E}_{\text{enc}}$ and $\mathbf{E}_{\text{dec}}$ are positive by definition, hence Lemma C.2 allows us to express the two minimisations as:

The left task is `autoencoding`, so we have that `energy minimisation` refines `autoencoding`. For the other direction, we observe that `autoencoding` refines the right task by postcomposing both sides with enc, and as the left and right tasks together are equivalent to `energy minimisation`, we have the claim. $\square$

## C.2 Proof of Proposition 3.2

`CycleGANs` (Example 2.6) solve a similar task as the `manipulator`, translating between two distributions. In fact, with the additional regularisation terms, the `strong manipulator` (see Appendix B) is a refinement of `CycleGANs`, giving us more guarantees by avoiding certain failure cases. We can use the pattern language to show this formally. However, in both cases, the tasks are not perfectible, as, by design, it is impossible for the generator and the discriminator to have a loss of zero at the same time. Therefore, we have to generalise the definition of *refinement* for *partially perfectible tasks*.

**Definition C.4** (Partially perfectible task). A *partially perfectible task* $\{(f_i, g_i, \mathcal{X}_i, \mathfrak{p}_i)\}_i$ with learnable functions $\Sigma_l$ is a compound task with excluded learners $E \subseteq \Sigma_l$ such that:

1. no two $e \in E$ share atomic tasks, i.e. $\forall e, e' \in E.\{(f_i, g_i, \mathcal{X}_i, \mathfrak{p}_i) | e \in f_i \vee e \in g_i\} \cap \{(f_i, g_i, \mathcal{X}_i, \mathfrak{p}_i) | e' \in f_i \vee e' \in g_i\} = \emptyset$ and

2. all tasks not involving learners from $E$ are perfectible, i.e. $\exists \pi \in \mathfrak{p}. \forall x \in X. f_{i;\pi}(x) = g_{i;\pi}(x)$

---

[4]When the function space containing $f$ and $g$ is large, the edge case where $f$ and $g$ are nonconstant and $f = -g$ is negligible. Since we are concerned with behaviour in the computational limit, this is an acceptable assumption for our purposes.

Importantly, there does not need to exist a perfect solution for all tasks. As condition (1) explicitly forbids excluded learners to share tasks, we do not need to make restrictions on their composite behaviour. An example of a partially perfectible task is `CycleGAN` with excluded learners $\text{dsc}_1, \text{dsc}_2$. Assuming an appropriate data distribution (Wu et al., 2024), the `autoencoding` tasks, i.e. the reconstruction losses, are perfectible. However, the generator-discriminator tasks are not. `GAN` with excluded learner `dsc` is another example of a partially perfectible tasks. In this case no tasks are required to be perfectible, as all of them involve the discriminator.

When we have partially perfectible tasks, we have to define what we mean by an optimal solution.

**Definition C.5** (Optimal partially pefect solution). Let $\Phi = \{(f_i, g_i, \mathcal{X}_i, \mathfrak{p}_i)\}_i$ be a partially perfectible tasks with learners $\Sigma$ and excluded learners $E \subseteq \Sigma$. Then an *optimal partially perfect solution* for the learners $L = \Sigma \setminus E$ is, if it exists, a set of parameters $\pi_L \in \mathfrak{p}_L$ such that for all $\alpha_L, \mathbf{D}_\phi$ they perfect the perfectible tasks:

$$\forall \{(f_i, g_i, \mathcal{X}_i, \mathfrak{p}_i) \in \Phi \,|\, \forall e \in E. e \notin f_i \wedge e \notin g_i\}. \forall x \in \mathcal{X}_i. f_{i;\pi_L}(x) = g_{i;\pi_L}(x)$$

and for all $l \in L$ and all tasks $\Phi_l$ with para($l$) $\cap \, \mathfrak{p}_i \neq \emptyset$:

$$\alpha_l(\{ \underset{x \sim \mathcal{X}_i}{\mathbb{E}} (\mathbf{D}_\phi(\text{sys}_{\phi;(\pi_L \cup \mathfrak{p}_E(\pi_L))}(x), \text{spec}_{\phi;\pi_L \cup \mathfrak{p}_E(\pi_L)}(x))) \,|\, \phi \in \Phi_l\})$$

is minimal over all possible parameter combinations, where

$$\mathfrak{p}_E(\pi_L) = \bigcup_{e \in E} \inf_{\pi_e \in \mathfrak{p}_e} (\alpha_e(\{ \underset{x \sim \mathcal{X}_i}{\mathbb{E}} (\mathbf{D}_\psi(\text{sys}_{\psi;(\pi_L \cup \pi_e)}(x), \text{spec}_{\psi;(\pi_L \cup \pi_e)}(x))) \,|\, \psi \in \Phi_e\}))$$

In words, we consider a set of parameters *optimal partially perfect*, if they are perfect for the perfectible tasks and have a minimal objective function for the non-perfectible tasks, even if the excluded learners only optimise for themselves. This may not always exist, either because there is no solution that is optimal for all $\alpha_L, \mathbf{D}_\phi$ or as there may be more optimal solutions for the non-perfectible tasks that do not satisfy the perfectible tasks.

Given this, we can generalise the definition of *refinement* for partially perfectible tasks.

**Definition C.6** (Optimal refinement of partially-perfectible tasks). Given two partially-perfectible tasks $\Psi, \Phi$ with excluded learners $E_\Psi, E_\Phi$, we say that $\Psi$ optimally refines $\Phi$ if optimal partially perfect solutions for $\Sigma_{l_\Psi} \setminus E_\Psi$ can be composed to form optimal partially perfect solutions for $\Sigma_{l_\Phi} \setminus A_\Phi$.

For $A_\Psi, A_\Phi = \emptyset$, optimal refinement is equivalent to refinement.

To show that `manipulation` optimally refines `cycleGAN`, we need one assumption: we assume that the measure of statistical divergence and compound function have been chosen such that a generator in a generative-adversarial setting is optimal if and only if its output distribution is equal to the original distribution. As the goal of such a generative setting is to approximate the original distribution, this is quite a natural assumption. One possible choice of measure of statistical divergence and compound function is given by Goodfellow et al. (Goodfellow et al., 2020, Theorem 1) and has been proven to fulfil this assumption.

**Proposition C.7** (Manipulation vs. Style-transfer). Given appropriately chosen measure of statistical divergence and compound functions, the `strong manipulator` with excluded learner `dsc` optimally refines the `CycleGAN` with excluded learners $\text{dsc}_1, \text{dsc}_2$.

*Proof.* Let $\mathcal{X}_1, \mathcal{X}_2$ be two distributions over the same type. We create $\mathcal{X} = (x, i)$ for $x \in \mathcal{X}_i$ where the label $i$ indicates the distribution the data point came from. Assuming that we have a `strong manipulator` (get, put, disc) with an optimal partially perfect put, get, we can construct optimal partially perfect generators $G_1$ and $G_2$ for `CycleGAN`.

For $i \in 0, 1, j = 1 - i$, we define:

First, we show that the perfectible tasks are indeed perfected, i.e. that both the autoencoder tasks are fulfilled. We have:

Next, we will show that these generators are indeed globally optimal. By assumption, a generator is optimal if and only if its output distribution is indistinguishable from the training distribution. Thus, assuming that the `manipulator` is optimal with respect to the generative-adversarial tasks, `put` output distribution approaches $\mathcal{X} = (x, i)$ for $x \in \mathcal{X}_i$. But by PUTGET, we have:

Therefore, by CLASSIFY, $G_1$ and $G_2$ can only return values that are akin to values in $\mathcal{X}_1$ and $\mathcal{X}_2$ respectively. As these two labels make up the entirety of $\mathcal{X}$, $G_1$'s and $G_2$'s output distributions are indistinguishable from $\mathcal{X}_1$ and $\mathcal{X}_2$ respectively. Therefore, they are indeed optimal generators for their respective distribution.

But then we have shown that $G_1$ and $G_2$ are indeed optimal solutions to the `CycleGAN` pattern and therefore that the concept manipulator is a specialisation of the CycleGAN[5]. □

In turn, the `CycleGAN`, however, is not a refinement of the `strong manipulator`. This means there exist solutions to `CycleGAN`, which violate `strong manipulator`. In these image translation tasks, we expect the translators to change as little as possible to go from one distribution to the other, i.e. preserving as much information from the original as possible. However, the generators of the `CycleGAN` could, for example, flip the images horizontally. As the `autoencoder` tasks have an even number of generators on each side, this would be a 'perfect solution' to the outlined task, yet not the behaviour we would want or expect. In contrast, the PUTPUT rule of the `strong manipulator` does not allow this behaviour. As such, the `strong manipulator` gives us more guarantees than the `CycleGAN`.

Despite the additional guarantees, the `strong manipulator` does not guarantee that it indeed only changes as little as necessary to go from one distribution to the other. This can be seen when considering a limited toy scenario: the data has the four objects *circle*, *square*, *red* and *green*. There is no unique mapping between shapes and colours. Without further information, it is therefore completely impossible to say what it would mean to change as little as possible to go from shapes to colours. When considering this toy scenario, it becomes obvious that, without specifying further restrictions on all remaining attributes, it is impossible to know which mapping is correct. Yet, despite these theoretical concerns, in practice, even the weaker `manipulator` often converges to the desired behaviour, similar to `CycleGAN`s, which have even fewer guarantees.

### C.3 PROOF OF PROPOSITION 3.3

**Definition C.8** (Balanced entropy of an attribute). Given a distribution of data $\mathcal{D}$ on space $D$, we say that a distribution $\mathcal{A}$ over space $A$ represents an *entropy-balanced attribute* of $\mathcal{D}$ if there exists a complement type $C$ with distribution $\mathcal{C}$ such that we have the equality in distributions $\mathcal{D} = \mathcal{A} \times \mathcal{C}$ up an isomorphism of the underlying spaces $D \simeq (A \times C)$.

---

[5]In (Zhu et al., 2017), when doing style transfer, they additionally add the *identity loss* which enforces that generating an image in $X_i$ given an image from $X_1$ should return the identity. This perfectible task is also guaranteed by the `strong manipulator` by `cls` and `GetPut`

**Task C.9** (`entropy balancer`). This task is `autoencoder` with a regularisation condition on the `encoder` to be satisfied in distribution.

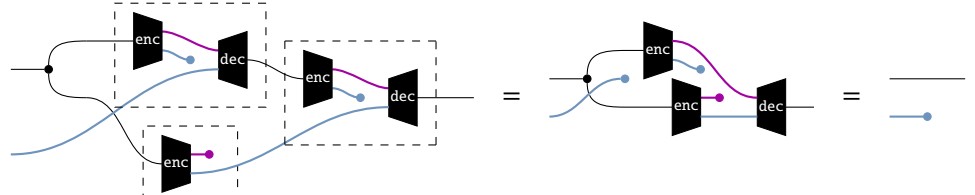

**Lemma C.10** (Perfect-task representation of equal entropy). Balanced entropy in the attributes is equivalent to the perfectability of `entropy balancer`.

*Proof.* If there is balanced entropy in the attributes, Task C.9 may be satisfied up to identity by the `enc` and `dec` witnessing the isomorphism $D \simeq (A \times C)$ that realises $\mathcal{D} = \mathcal{A} \times \mathcal{C}$. Conversely, if the tasks above are equalities, we recover the definition of balanced entropy. $\square$

Contextually, we will assume that distributions have full support over spaces; this can always morally be the case by restrictions to subspaces. This assumption strengthens Lemma C.10 to yield the following corollary.

**Corollary C.11.** If `entropy balancer` is perfect, `enc` and `dec` witness an isomorphism $D \simeq (A \times C)$.

**Lemma C.12** (Hardcoding latent-spaces). Given a perfected `entropy balancer`, we may construct the following composites, which are the `put` and `get` of a `strong manipulator`[6].

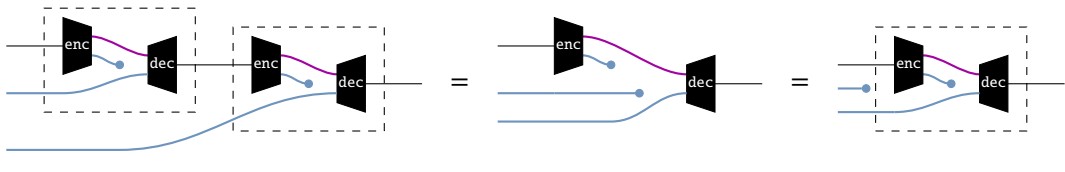

*Proof.* First, we argue that the lens laws are satisfied. PutGet, GetPut, Undoability, and PutPut follow from pure diagrammatic reasoning and applying Corollary C.11. For PutGet:

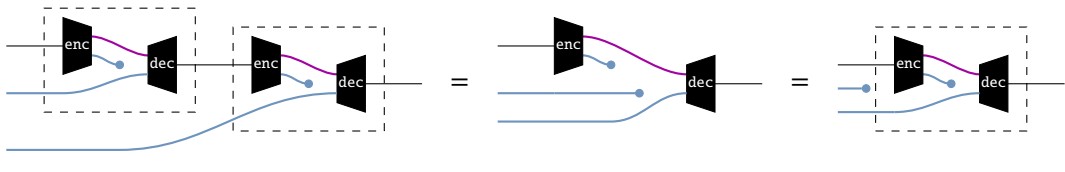

(Corollary)           (Delete naturality)

GetPut:

(copy naturality)           (copy-delete counitality)   (`autoencoder`)

Undoability:

(Corollary)           (GetPut)

PutPut:

(Corollary)

---

[6]modulo CLASSIFY, which in this setting is trivially obtained as we assume the attribute is derived from `get`.

By construction, putting $a$ by independently sampling the marginal on $A$ is indistinguishable in distribution from the original distribution, hence the GAN laws of strong manipulator are optimally satisfied, and we are done. □

**Definition C.13** (Bayesian Inversion in Markov Categories)**.** In a Markov category, the Bayesian inversion (Cho & Jacobs, 2019; nLab authors, 2024a) of a stochastic map $f : X \to Y$ with respect to a distribution $\mathcal{P}$ on $X$ is a stochastic map $f^\dagger : Y \to X$ such that, in distribution:

$$\mathcal{P} \vdash X \boxed{f} Y \bullet \boxed{f^\dagger} X \underset{Y}{\phantom{x}} \quad \underset{\text{(in dist.)}}{=} \quad \mathcal{P} \vdash \begin{matrix} X \\ X \boxed{f} Y \end{matrix}$$

**Proposition C.14** (strong manipulator as Bayesian inversion)**.** If a discriminative classifier cls : $D \to A$ induces a balanced attribute cls($\mathcal{D}$) over $A$ with respect to $D$, there exists a strong manipulator for which the put induces the Bayesian inversion cls$^\dagger$ : $A \to D$.

*Proof.* We show that, under our premises, the put composed with an independent copy of the data source $\mathcal{X}$ is the Bayesian inversion of cls.

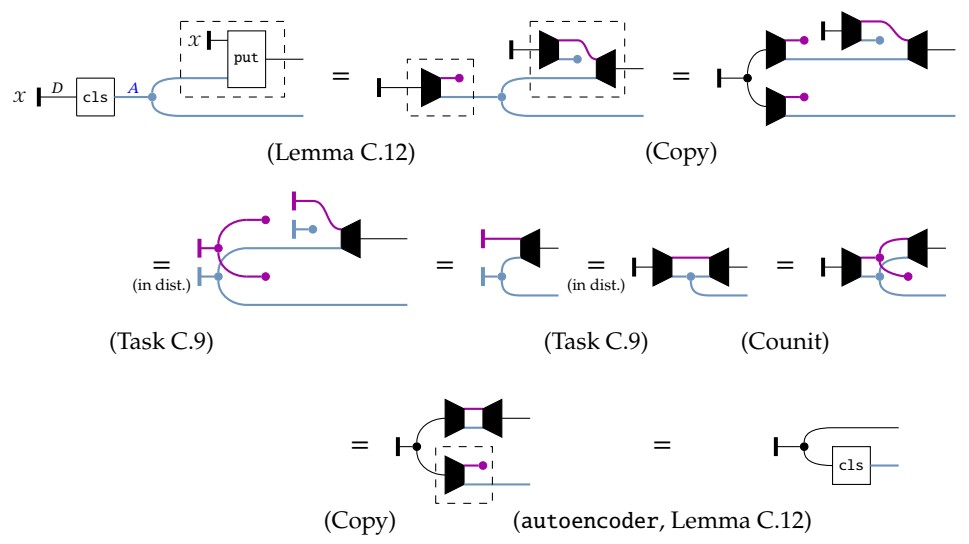

(Lemma C.12)    (Copy)

(Task C.9)    (Task C.9)    (Counit)

(Copy)    (autoencoder, Lemma C.12)

□

# D EXPERIMENT DETAILS

## D.1 SPRITEWORLD

For the Spriteworld experiment, we procedurally generate 32x32 images containing a single shape with the following attributes:

| Attributes | Possible Values |
|---|---|
| Shape | { Ellipse, Rectangle, Triangle } |
| Hue | { Red : 0±8, Green : 85±8, Blue : 170±8 } |
| Saturation | 64-255 |
| Value | 64-255 |
| Background Color | Black |
| Width & Height | 5-27 |
| X & Y position | 5-27 |

Only the first two attributes, shape and hue, are changed in the manipulation task, but all unchanged properties are intended to be preserved by the transformation. We use an autoencoder with a CNN/DCNN architecture to embed each image into a latent space:

| Parameter | Value |
|---|---|
| Latent Size | 32 |
| Layers | 4 |
| Hidden Channels | 64 |
| Kernel Size | 5x5 |
| Stride | 2 |
| Activation Function | LeakyReLU(0.1) followed by BatchNorm |

We train separate `get`/`put` models for each of the three concepts: shape, colour, blue-circleness. Each model uses the encoder of the autoencoder to embed input images into latent space, and only sees the labels for the particular attribute it is manipulating.

For shape and colour, the `get` model uses a linear classifier from the latent space (of size 32) to 3 output logit values, one for each possible value. The `put` model maps a one-hot vector of the input value to a vector in latent space that is added to the embedding. This new embedding is then decoded by the autoencoder.

For blue-circleness, the `get` model uses a linear classifier from the latent space to a single output value from zero to one (we do not use a sigmoid output layer to restrict the output). The `put` model uses a complement of size 8. It concatenates the one-hot value vector with the image embedding and the complement vector (using a default trainable complement vector if one is not provided) and passes that through a linear layer to get a new embedding vector (which is then decoded) and complement vector.

In total, these models contain 644,130 parameters. All models are trained simultaneously according to the `autoencoding` and `manipulation` rules, along with PutPut. At each step, a batch of images is generated, along with four batches of random values, containing random labels for the shape and colour manipulators, and random real numbers for the blue-circleness manipulator, uniformly sampled from $[-0.1, 1.1]$ and then clamped to $[0, 1]$. The loss function is a weighted sum of the losses from each atomic task in order to balance the signal from the image loss with the signal from the classifier loss:

| Hyper-parameter | Value |
|---|---|
| Steps | 100,000 |
| Batch Size | 512 |
| Optimiser | AdamW |
| Learning Rate | $10^{-3}$ |
| Weight Decay | $10^{-2}$ |
| Gradient Clipping | 1 (element-wise) |
| Image Loss | $L_2 + 0.25 \cdot L_1$ |
| Discrete Value Loss | Binary cross-entropy |
| Continuous Value Loss | Mean squared error |
| Seed | 0 |

| Task | Weight |
|---|---|
| AUTOENCODING | 100 |
| GETPUT | 1 |
| PUTPUT | 1 |
| UNDO | 10 |
| PUTGET | |
| (blue-circleness) | 10 |
| (shape and colour) | 1 |
| CLASSIFICATION | |
| (blue-circleness) | 10 |
| (shape and colour) | 1 |

## D.2 FACES

For the faces experiment, we use the *CelebFaces Attributes* dataset (Liu et al., 2015), with an off-the-shelf data augmentation method called "TrivialAugment" (Müller & Hutter, 2021). Again, we use an autoencoder with a CNN/DCNN architecture to embed each image:

| Parameter | Value |
|---|---|
| Latent Size | 128 |
| Layers | 5 |
| Hidden Channels | 8, 16, 32, 64, 128 |
| Kernel Size | 5 |
| Stride | 2 |
| Activation Function | LeakyReLU(0.1) followed by BatchNorm |

We train linear `get`/`put` models for the binary concept of "Smiling", resulting in a total of 1,071,749 parameters. The loss function is a weighted sum of the losses from each atomic task:

| Hyper-parameter | Value |
|---|---|
| Steps | 100,000 |
| Batch Size | 64 |
| Optimiser | AdamW |
| Learning Rate | $10^{-3}$ |
| Weight Decay | $10^{-2}$ |
| Gradient Clipping | 1 (element-wise) |
| Image Loss | $L_2 + 0.2 \cdot L_1 + SSIM$ |
| Value Loss | Binary cross-entropy |
| Seed | 0 |

| Task | Weight |
|---|---|
| AUTOENCODING | 10 |
| GETPUT | 1 |
| PUTPUT | 1 |
| UNDO | 1 |
| PUTGET | 1 |
| CLASSIFICATION | 1 |

### D.3 MNIST

We trained the `manipulator` pattern on the MNIST dataset, using the digit label as the property. The `get` operated directly on images, while `put` was trained to act on the latent space of an autoencoder, as in option (1) of Section 3.3. The images are input as $28 \times 28$ matrices, flattened to 784-dimensional vectors, and the labels are provided as 10-dimensional vectors with one-hot encoding. All of the components were structured as multilayer perceptrons. The hyperparameters are given below:

| | enc | dec | put | get |
|---|---|---|---|---|
| Input Dimension | $784 = 28 \times 28$ | 32 | $42 = 32 + 10$ | $784 = 28 \times 28$ |
| Output Dimension | 32 | $784 = 28 \times 28$ | 32 | 10 |
| Hidden Dimensions | $\{128, 128, 64\}$ | $\{64, 128, 128\}$ | $\{128, 128, 128\}$ | $\{64, 64\}$ |
| Hidden Activations | ReLU | ReLU | ReLU | ReLU |
| Final Activation | Sigmoid | Sigmoid | Sigmoid | Softmax |

With these architectural components, we trained four tasks: (a) training `get` supervised, (b) training `get` given pre-trained `put`, `enc`, and `dec`, (c) training `put` given pre-trained `get'`, `enc`, and `dec`, and (d) training `get` to match a previous `get'`. These were trained using the `manipulation` rules, as well as PUTPUT, and additional regularization term we denote as ENTROPY. The loss function of ENTROPY is given by

$$\mathcal{L}_{\text{ENTROPY}} = \mathbb{E}[H(\texttt{get}(\texttt{enc}(x)))] - H(\mathbb{E}[\texttt{get}(\texttt{enc}(x))])$$

where $x$ is a batch of input images, $H(\cdot)$ is the entropy of a categorical distribution, and the expectation is approximated by the mean over each batch. The idea behind ENTROPY is to encourage the output of `get` to be well-distributed across labels (by maximizing the entropy of the mean distribution) but to be sure of each label (by minimizing the entropy for each specific input). For task (d), labels were generated for the CLASSIFY rule using `get'`. Each task is trained by minimizing a weighted linear combination of the rules. We give the hyperparameters, rule weights, and loss functions for each of these below.

| | (a) | | (b) | | (c) | | (d) | |
|---|---|---|---|---|---|---|---|---|
| Optimizer | Adam | | Adam | | Adam | | Adam | |
| Learning Rate | 0.001 | | 0.001 | | 0.0001 | | 0.001 | |
| Epochs | 20 | | 20 | | 20 | | 20 | |
| | *Weight* | *Loss* | *Weight* | *Loss* | *Weight* | *Loss* | *Weight* | *Loss* |
| CLASSIFY | 1 | CE | – | – | – | – | 1 | CE |
| PUTGET | – | – | 10 | CE | 10 | CE | – | – |
| GETPUT | – | – | 10 | L2 | 10 | L2 | – | – |
| PUTPUT | – | – | 10 | L2 | 10 | L2 | – | – |
| UNDOABILITY | – | – | 10 | L2 | 10 | L2 | – | – |
| ENTROPY | – | – | 1 | $\mathcal{L}_{\text{ENTROPY}}$ | – | – | – | – |

We also have an additional task (e) of training `enc` and `dec` unsupervised. This was done using the Adam optimizer, with a learning rate of 0.001 for 80 epochs. The reconstruction loss was given by

$$\mathcal{L}(x, \hat{x}) = \text{L2}(x, \hat{x}) + (1 - \text{SSIM}(x, \hat{x}))$$

where SSIM is the structure similarity image metric.

To produce Figure 2, an `enc`, `dec`, `put`, and `get` were trained using (a) → (e) → (c), followed by training (*c*) for an additional 40 epochs. A slightly larger (but still MLP-based) model, where both `put` and `get` act on the latent space of the autoencoder, was used to achieve better visual quality. The hyperparameters are detailed below:

| | enc | dec | put | get |
|---|---|---|---|---|
| Input Dimension | $784 = 28 \times 28$ | 32 | $42 = 32 + 10$ | 32 |
| Output Dimension | 32 | $784 = 28 \times 28$ | 32 | 10 |
| Hidden Dimensions | {128, 128, 128, 32, 32} | {32, 32, 128, 128, 128} | {256, 256} | {256} |
| Hidden Activations | ReLU | ReLU | ReLU | ReLU |
| Final Activation | Sigmoid | Sigmoid | Sigmoid | Softmax |

`enc`, `dec`, and `put` were used to manipulate six examples picked from the dataset, putting each of the ten classes onto each example. The examples were cherrypicked to provide maximum stylistic contrast across the sample but were not selected for maximum style transfer accuracy - a similar level was observed across the entire dataset. Code for all of these models, as well as the training schedules of tasks (a)-(e), are provided in the supplementary material.

# E  STACKS

Here we define tasks that obtain learned operators that implement the well-known data structure `stack`. There are two interacting operations: *push* and *pop*. By itself, the `stack` is not particularly interesting, as it can algorithmically be implemented rather easily, but a fully differentiable and automatically learned `stack` may be useful for other downstream tasks.

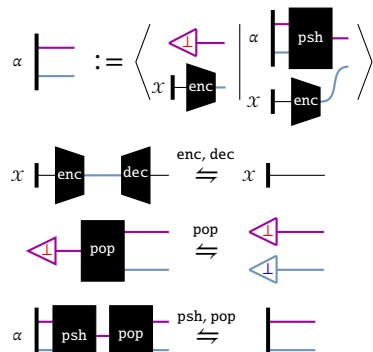

Given a data distribution $\mathcal{X}$ over $X$ and a pretrained autoencoder (enc: $X \to \textbf{LAT}$, dec: $\textbf{LAT} \to X$), a `stack` consists of a *empty stack* $\bot : \star \to S$, a push operation $psh : S \times X \to S$ and a pop operation $pop : S \to S \times X$. We recursively define the distribution:

$$\alpha := \bot \otimes \text{enc}(\mathcal{X}) \quad | \quad \text{psh}(\alpha) \otimes \text{enc}(\mathcal{X})$$

meaning $\alpha$ is a stack of arbitrary size and an encoded element from $\mathcal{X}$.

Then `psh` and `pop` have to obey the rules on the left.

Experimentally in the Spriteworld setting, we observe that the stack works exactly as expected. The stack space $S$ has to be $n$ times larger than the latent space **Lat** for the stack to be able to hold up to $n$ items. At the moment, we do not observe any additional information compression in this basic setup. The experiment (Figure 7) uses an autoencoder architecture for processing $32 \times 32$ RGB images, with a latent space size of 16 dimensions. The encoder consists of four convolutional layers, each with 64 channels and a channel multiplier of 1. Additionally, a stack of 64 latent features is processed through an MLP with 256 hidden units. Training is performed on a GPU (if available), with a batch size of 64, learning rate of $1 \times 10^{-4}$, weight decay of $1 \times 10^{-2}$, and gradient clipping at 1. The model trains for 100,000 steps, logging every 10,000 steps. Input images are converted to tensors and scaled to floating-point precision, and a fixed random seed of 0 ensures reproducibility.

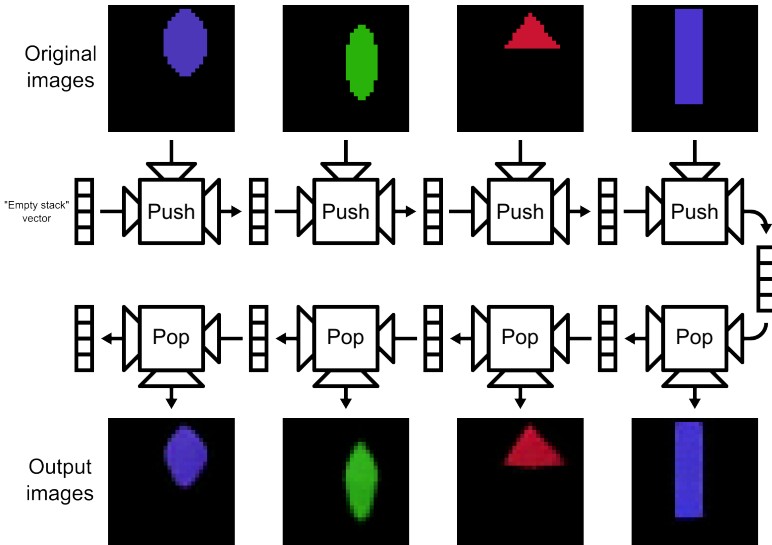

Figure 7: In this example, we train a stack (alongside an autoencoder) to store the latent vectors of Spriteworld shapes. With an image latent size 16 and stack vector size 64, it is able to retain information to faithfully restore up to 4 shapes.

## F    EXAMINING MANIPULATION IN COMPLEX DOMAINS

### F.1    MANIPULATION FOR TEXT SENTIMENT

We attempted to fine-tune an extant strong solution for text-sentiment modification by additionally imposing the constraints of `manipulation` on top of the original objective function. Our findings suggest that **additionally imposing the constraints of `manipulation` on architectures that are already performant does not make an appreciable difference**

(Table 1). We pretrained the Blind Generative Style Transformer (**B-GST**) model of (Sudhakar et al., 2019) which takes in the non-stylistic components of a sentence and the target sentiment, and outputs the sentence generated in the target style. This was done until we achieved baselines higher than the original ones reported by the authors of the model. Afterwards, we continued training under three different conditions: (1) resuming training with only the original objective, (2) only using objective functions from `manipulation`, and (3) using both. For (1) and (3), we reused the same objective in the original paper. In all experiments, we used the YELP dataset used by (Li et al., 2018), reusing the same train-dev-test split they used. It consists of 270K positive and 180K negative sentences for the training set, 2000 sentences each for the dev set, and 500 sentences each for the test set. Furthermore, we used the human gold standard references they provided for their test set. **B-GST** uses a sequence length of 512, 12 attention blocks each with 12 attention heads. We used 768-dimensional internal states (keys, queries, values, word embeddings, positional embeddings). We tokenized the input text using Byte-Pair Encoding (BPE).

We used the same input autoencoding and output decoding used in (Sudhakar et al., 2019) across all experiments. For the `get` of the `manipulation` task, we used the PyTorch version of the pretrained Transformer by HuggingFace, which uses the OpenAI GPT model pretrained by (Radford & Narasimhan, 2018) on the BookCorpus dataset which contains over 7000 books with approximately 800M words. We trained it on a sentiment classification task using the YELP dataset reaching 98% accuracy on the test set. The `get` was fixed for the entire duration of training conditions (2) and (3) above. For the `put` of `manipulation`, we used the **B-GST** model to generate text with a specified sentiment. This was a computational bottleneck for training conditions (2) and (3) as autoregressive decoding is required to generate model inputs for PUTGET and UNDOABILITY in `manipulation`. We used 'teacher forcing' or 'guided approach' (Bengio et al., 2015; Williams & Zipser, 1989) whenever we computed the reconstruction loss of `put`. Additionally, for training conditions (2) and (3), we only used PUTGET, GETPUT, and UNDOABILITY from `manipulation`. We used a weighted sum of the losses computed for each of these and the original reconstruction loss if present - the weights can be considered as training hyperparameters. For (2), we used (PUTGET=5, GETPUT=20, UNDOABILITY=20), while for (3), we used (PUTGET=5, GETPUT=10, UNDOABILITY=25, B-GST=30). Code for all of the models, training schedules, and hyperparameter values for training conditions (1)-(3) are also provided in the supplementary material.

| MODEL | GLEU | BLEU$_{\text{SRC}}$ | BLEU$_{\text{REF}}$ | ACC (FASTTEXT) |
|---|---|---|---|---|
| B-GST-pretrained | **11.869** | 74.563 | 52.770 | 84.6 |
| B-GST-only | 11.426 | **74.876** | 52.549 | **85.7** |
| manipulation-only | 11.712 | 74.428 | 52.646 | 84.1 |
| B-GST+manipulator | 11.338 | 74.608 | **52.836** | 85.1 |
| Human Reference | 100.00 | 58.158 | 100.00 | 67.6 |

Table 1: We pretrained the Blind Generative Style Transformer (**B-GST**) model (Sudhakar et al., 2019) based on the `Delete-Retrieve-Generate` (Li et al., 2018) framework for sentiment modification until we recovered higher baselines than reported by the authors of the model (GLEU=11.6, BLEU$_{\text{SRC}}$=71.0), and we continued training in three different conditions: (1) keeping the original objective, (2) only using objective functions obtained from `manipulation`, and (3) using both. We report no statistically significant differences in scores, even under continued training. Below we report the results of training conditions (1) and (2) for an additional epoch, and (3) for two epochs (details in Appendix F).

### F.2 CHARACTERISING MANIPULATION AS GENERATIVE CLASSIFICATION

Training `manipulation` autoregressively (e.g. when instantiating the learners as transformers for sequential data) is slow due to autoregressing twice for PUTPUT and UNDOABILITY. Moreover, our attempts to autoregressively manipulate the sentiment of IMDB reviews often resulted in a form of posterior collapse where `put` ignored the attribute and behaved as the identity function on text. Conceptually, this is because the identity function satisfies

GETPUT, PUTPUT and UNDOABILITY, and while the identity fails on PUTGET, failing on one component of the combined loss function does not provide a strong enough incentive to move away from the identity in parameter-space.

Notably, these shortcomings mirror that of VAEs, which also suffer from posterior collapse (Bond-Taylor et al., 2022) in highly structured domains such as video (Babaeizadeh et al., 2018) and text (Bowman et al., 2016). This suggested to us that **puts may be generative classifiers, which could potentially explain why mode collapse was occurring in complex domains.** Borrowing terminology from (Ng & Jordan, 2001), classifiers are *discriminative* if they seek to learn the conditional distribution $p(a|d)$ of attributes given data (as in the `classification` pattern), and otherwise they are *generative* if they seek to learn the joint distribution $p(d, a)$ (as, for example, a VAE). The tradeoffs between the two types are well studied, e.g. performance-wise, generative models may converge faster with limited data, but discriminative models often achieve lower asymptotic error, and it is well known that learning generative models is harder (Vapnik, 1998).

**As generative classifiers, `puts` in the `manipulation` task appear to approximate the Bayesian inverse of a discriminative classifier `cls`, approximately as well as VAEs do.** To demonstrate this, we tried three ways to train an "informationally identical" `cls'` given an initial `cls`, provided access to unlabelled data to obtain a distribution of pairs $(d, \text{cls}(d))$ by: (a) directly training `cls` by `classification`, (b) training a generative classifier, in our case a VAE, and (c) training `manipulation` around `cls-as-get` to obtain a `put`, and then train `cls'` to satisfy the tasks of `manipulation` except for CLASSIFY. Repeating this process several times, we would expect to see some loss of accuracy due to imperfect Bayesian inversion. Indeed, we see in Figure 8 that (b) and (c) have similar decays in accuracy, indicating that `manipulation` and VAEs have similar performance in this case. This experiment was performed using the trained components obtained from the MNIST experiment (Section D.3), and in addition to the tasks (a-e) we (f) trained a VAE to learn the joint distribution of images and labels produced by `get'`, and we (g) trained a `get` supervised using labels and images generated from the VAE. The VAE encoder and decoder are also based on multilayer perceptrons - each is comprised of an MLP trunk and two linear heads for generating the means and log-variances of the latent space, or the image and labels, respectively. The latent space is comprised of independent normally distributed variables as in (Kingma & Welling, 2022), and is sampled using the standard reparameterization trick. The hyperparameters of the architecture are given below:

|  | VAE Encoder | VAE Decoder |
|---|---|---|
| Input Dimension | $794 = 28 \times 28 + 10$ | 32 |
| Hidden Dimensions | $\{128, 128, 64\}$ | $\{64, 128, 128\}$ |
| Hidden Activations | ReLU | ReLU |
| Head 1 Dimension | 32 | $784 = 28 \times 28$ |
| Head 1 Activation | – | Sigmoid |
| Head 2 Dimension | 32 | 10 |
| Head 2 Activation | – | Softmax |

Three loss functions were used for tasks (f) and (g) — the reconstruction loss of the autoencoder, which can be separated into a label loss and an image loss, the K-L divergence regularization term $\mathcal{L}_{KL}$ of the VAE (Kingma & Welling, 2022), and the CLASSIFY loss of `get`. The training hyperparameters are given as follows:

|  | (f) | | (g) | |
|---|---|---|---|---|
| Optimizer | Adam | | Adam | |
| Learning Rate | 0.001 | | 0.001 | |
| Epochs | 40 | | 20 | |
|  | *Weight* | *Loss* | *Weight* | *Loss* |
| CLASSIFY | – | – | 1 | CE |
| Image Reconstruction | 100 | L2 | – | – |
| Label Reconstruction | 1 | CE | – | – |
| K-L Divergence | 0.5 | $\mathcal{L}_{KL}$ | – | – |

In order to evaluate the three methods, we trained the tasks in the following order. At each step, the component being trained (e.g. `get`, `put`, etc) was initialized randomly (the previous weights were discarded). Measurements of the test accuracy were made after each (a), (b), (d), or (g) training run, and used to produce Figure 8.

$$\texttt{get} \rightarrow \texttt{get'} \implies (a) \rightarrow (d) \rightarrow (d) \rightarrow \cdots \rightarrow (d)$$
$$\texttt{get} \rightarrow \texttt{put} \rightarrow \texttt{get'} \implies (a) \rightarrow (e) \rightarrow (c) \rightarrow (b) \rightarrow (e) \rightarrow (c) \rightarrow (b) \rightarrow \cdots \rightarrow (b)$$
$$\texttt{get} \rightarrow \texttt{VAE} \rightarrow \texttt{get'} \implies (a) \rightarrow (f) \rightarrow (g) \rightarrow (f) \rightarrow (g) \rightarrow \cdots \rightarrow (g)$$

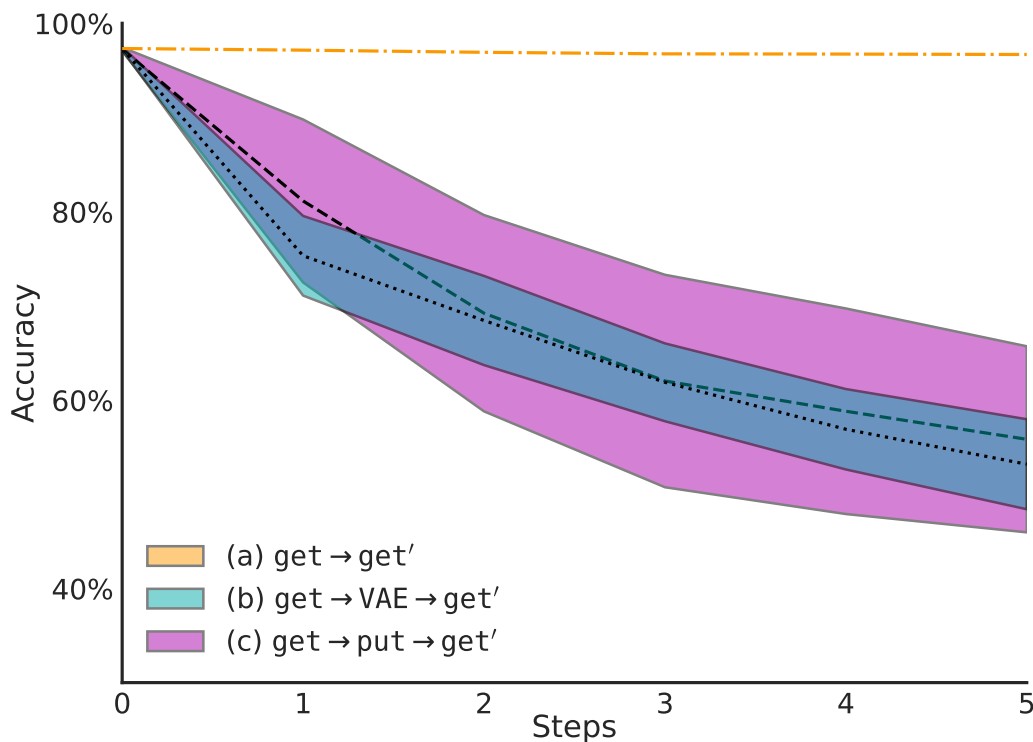

Figure 8: We tried three ways to train an "informationally identical" `cls'` given an initial `cls` - depicted are the results of training successive MNIST classifiers using methods (a), (b) and (c) given above. 'steps' refers to the number of times this process was repeated. We observe that the degradation of accuracy is approximately the same for both (b) and (c), **which we consider evidence that `manipulator` and `VAEs` have similar performance characteristics**. Both models had roughly the same number of parameters (300K) and were based on the same MLP architecture. We ran 20 repetitions of each method, the shaded regions represent one standard deviation (method (a) had a standard deviation of less than 1%).

