# OpenReview forum: "A Pattern Language for Machine Learning Tasks"
_ICLR.cc/2025/Conference — Submitted to ICLR 2025_

### Official Review · Reviewer_GYtH · 2024-11-02

**Soundness:** 2
**Presentation:** 2
**Contribution:** 2
**Rating:** 3
**Confidence:** 3

**Summary:**

This paper proposes a "pattern language", which treats machine learning tasks as constraints that specify desired equivalences.
Some commonly used tasks (e.g. classification, autoencoding, GAN) are called "patterns", borrowing terminology from software engineering.

The pattern language abstracts away implementation details (e.g. architecture and training), by treating neural networks are universal approximators that can perfectly solve the task.

The paper uses a task called "manipulation" as a proof-of-concept. The paper shows results on manipulating attributes, such as interpolating between shapes and colors of synthetic images, and smile/no-smile on CelebA.

**Strengths:**

This proposed pattern language with string diagrams provides a type of high-level abstraction of the machine learning pipeline, which can be helpful for conceptual understanding and can help provide intuition for someone who is entering the field.

**Weaknesses:**

Although providing a clear mental picture, the benefit of this framework is not clear to me.
- I'm not sure how the string diagrams can help advance theoretical/mathematical understanding; it would be helpful if the paper could provide some concrete examples.
- I don't see how the proposed pattern language improves empirical results either. For example, my understanding is that composing tasks is equivalent to adding regularization terms, which has been commonly used in practice.

I'm also concerned about the audience. I feel that the intended audience of the paper is not machine learning researchers/practitioners, but rather people with a software engineering or programming language background.
- This is partially reflected in the choices of terminology and notation. For example, this paper uses $\mathbb{P}$ to denote datatypes (e.g. of trainable parameters), which is an uncommon choice since $\mathbb{P}$ is usually used for probability-related quantities. Please see the clarification questions below for more examples. The concepts such as "symmetric lenses" are unfamiliar to most ML audience and would require more explanation.

**Questions:**

Regarding the significance of the results, could you explain how the proposed pattern language can help with theoretical understanding or empirical advances?
- For example, it's unclear to me how "(composite) tasks" differ from multitask learning or adding regularization terms.

Some clarification questions:
- For specialised tasks, "structural inductive biases" refer to parameterization?
- line 188: "energy minimisation architecture": is "architecture" the same as  "pattern", i.e. a particular composition of tasks?
- Prop 3.4: the meaning of "balanced" in "balanced attribute" has not been explained (until the appendix).

---

> ### Author Response · Authors · 2024-11-17
>
> We thank you for your thoughtful feedback, particularly regarding the paper's positioning for the ML community. While we agree that better connecting to ML terminology would improve accessibility, we believe the substantive benefits for ML practitioners are stronger than your review suggests, and we welcome the opportunity to clarify this.
>
> Before we address your specific questions, your key concerns appear to be:
> 1. How our framework differs from standard regularization/multitask learning
> 2. Benefits for ML practitioners beyond conceptual understanding
>
> Regarding the distinction from regularization/multitask learning:
> As you observe, tasks can express both of these (modulo choices of stat. divs. and weighting), but tasks can additionally express "multi-learner multi-task" learning; rather than simply adding loss terms to a single model, our framework guides the principled construction of modular objectives for distinct learners, which work together to solve problems. While that in itself is not new -- notably GANs work this way, but the discriminator is 'thrown away' at the end of training -- we seek to capture the phenomenon in generality. As demonstrated in Section 3.1 (ln 276), this perspective yields concrete benefits - we obtain a generative model that:
> - Works without supervised pairs and deterministically (cf. VAEs)
> - Trains stably without adversarial objectives (cf. GANs)
> - Allows targeted architectural improvements while preserving formally specified behaviours (Section 3.3's linear put)
>
> Concerning benefits beyond conceptual understanding:
> Theoretically, we have Proposition 2.10 which formally relates energy minimisation (and therefore, unsupervised learning approaches such as k-means) to autoencoding, and proposition 3.4 which relates the behaviour of strong manipulation to bayesian inversion. The proofs of these propositions in Appendix B is diagrammatic, making full use of the soundness and completeness of such diagrams for cartesian monoidal categories; if nothing else, it is a different way to do mathematics.
>
> Practically, echoing our responses to reviewer 2Po6 and mqHa, the benefits are downstream of a systematic recipe for developing new ML training objectives, which we will state explicitly in the paper if you find the perspective helpful:
> 1. Begin with an equational specification of desired behaviour - formal constraints that captures how components should interact. While there's no universal recipe for converting desired behaviors into equations, we provide a foundation of well-understood patterns (Section 2) that can be analyzed and composed to derive novel objectives - as demonstrated by our manipulation task. As novel examples, Section 3.1 specifies what properties a well-behaved attribute editor should satisfy, and Appendix E specifies how to translate algebraic conditions governing push and pop into a neural stack.
> 2. Replace abstract functions with learnable components (e.g., neural networks). The framework is agnostic to specific architectural choices, treating learners as universal approximators (detailed in Appendix A.2).
> 3. Transform equations into trainable objectives by choosing appropriate statistical divergences and combining weights (subject to hyperparameter search). These choices determine the specific overall loss function to be minimised. The resulting trained components will exhibit the specified behavior (in the computational limit) because the learning process finds smooth function approximations satisfying the constraints. As we then demonstrate, not only does this recipe work, the practical advantages we win are:
> 1. Systematic design of objectives (Section 3)
> 2. Formal guarantees about learned behavior (Props 3.3, 3.4)
> 3. Principled architectural modifications (Section 3.3)
> 4. Novel training approaches (manipulation task)
>
>
> Given the concrete benefits demonstrated in the paper, we believe our contribution is stronger than just providing a clear mental picture. We welcome your thoughts on whether these clarifications address your concerns about practical value.
>
> (specific questions to follow)

---

> > ### Author Response · Authors · 2024-11-17
> >
> > Specific questions:
> >
> > "Regarding the significance of the results, could you explain how the proposed pattern language can help with theoretical understanding or empirical advances? For example, it's unclear to me how '(composite) tasks' differ from multitask learning or adding regularization terms."
> >
> > Hopefully this has been adequately addressed above. We look forward to more clarifying discussions if necessary.
> >
> >
> >
> > "For specialised tasks, 'structural inductive biases' refer to parameterization?"
> >
> > Not exactly, but it is indeed our fault for not clarifying this term. In this case what we mean is "architectural specification", e.g. choosing between a fully-connected NN or a CNN. We will more explicitly list out the kinds of relationships between diagrams we consider throughout the paper.
> >
> >
> >
> > line 188: "energy minimisation architecture": is "architecture" the same as "pattern", i.e. a particular composition of tasks?
> >
> > Yes. Our apologies for the confusion, we just mean pattern here and we will clarify. Outside of this paper, the boundaries between "model", "architecture", and "objective" may be blurry; in this work we are primarily interested in objective functions on multiple learners.
> >
> >
> >
> > Prop 3.4: the meaning of "balanced" in "balanced attribute" has not been explained (until the appendix).
> >
> > We will signpost better that these terms are introduced in the appendix; we would prefer not to clutter the main body with these details for flow.

---

> > > ### Comment · Reviewer_GYtH · 2024-12-02
> > >
> > > Thank you for the clarifications!
> > >
> > > Re "structural inductive biases": by "parameterization" I was also referring to the architectural aspects; i.e. if we consider a neural net as a function, how is the function parameterized -- my bad for not being clear earlier (yet another example of confusing language!).

---

> > ### Comment · Reviewer_GYtH · 2024-12-02
> >
> > Thank you for the clarifications!
> > I'm clearer about the relationship to regularization/multitask learning, though I'm still not convinced about the benefits, especially about how the proposed framework can help with training stability.
> > I'd like to keep my score.

---

### Official Review · Reviewer_zN4b · 2024-11-04

**Soundness:** 3
**Presentation:** 3
**Contribution:** 3
**Rating:** 6
**Confidence:** 2

**Summary:**

This work takes an idealized view of model optimization to formalize "task" in terms of a diagram that captures the behavior objective of the model being trained. This formalism is intended to provide better understanding of tasks as well as their relationships to other tasks. Using this formalism, they provide diagrams for several well known algorithms (such as classifiers, gans, and auto encoders), as well as formulate a relationship between auto encoding and energy minimization. Using their specification framework (i.e., "language"), they construct a possibly novel task called "manipulation" and demonstrate its efficacy on toy tasks.

**Strengths:**

In general, the work is well written and easy to understand. The motivation of introducing the formal language for describing tasks makes sense and the applications are well delivered in a number of common or well-known objectives. Generally this sort of formalism could make it easier to relate different algorithms w.r.t. their target behavior and system specification. The novel training paradigm I believe is novel (manipulation), which is convincingly yielded from the formalism and its efficacy is well established through experiments.

**Weaknesses:**

Most of the issues are on clarification, for which I have added questions below.

Some of the symbols could be better presented, as I continuously had to refresh myself on what the numerous symbols meant while building an understanding of the formalism. Perhaps a table or some other sort of summary that helps the reader reference while looking through diagrams.

One major issue I have is it's unclear how the formalism specifically relates different algorithms w.r.t. the target behavior of the model. Is there a set of rules that allow the user to transform and reduce to a "unique" diagram that represents a group of tasks with similar target behaviors? Is uniqueness part of what this formalism brings (which makes finding relations easier), or can be never guarantee a unique representation of the task (e.g., there are equivalences which are not deterministically reducible to the same representation).

The end complaint here is for me the clearest demonstration of usefulness would be more concrete examples on how to relate / compare algorithms using operations on the language. Is this something the language provides? If not, are there reasons why it can at least help relate algorithms?

**Questions:**

1) Could you clarify what the role of the compound function, alpha, is? It appears to have something to do with the gradients of the parameters, but either way this should be made explicit or given an example in the work.
2) Could you clarify what the domain and codomain are? From what I can tell, these are the inputs and outputs of the diagrams (left to right), but this could be clearer, either way.
3) is "system" and "specification" synonymous with "(trained) behavior" and "target" (behavior)?
4) In Pattern 2.8 should there be a "y" in the domain of the specification?
5) why is "get" above the arrow for task 3.2 PutPut?
6) Is there a way through schematic manipulation to show equivalence between tasks? Eg between manipulation and strong manipulation it's implied they do similar things, but their diagrams are quite different. How would a practitioner understand that they are doing similar things? (perhaps some of this is in B2,3, but I didn't get a chance to look carefully at the Appendix)

---

> ### Author Response · Authors · 2024-11-17
>
> We deeply appreciate your insightful review, which cuts to the heart of the framework's potential. Your questions about algorithm relationships and diagram transformations are particularly valuable, as they highlight areas where we can strengthen the presentation to better serve the broader ML community.
>
> You raise three crucial conceptual questions, which we would like to address before your specific questions:
> 1. How to formally relate different algorithms through the framework
> 2. Rules for diagram transformation and equivalence
> 3. Clear demonstration of comparative insights
>
> Regarding algorithm relationships: As you've recognized, our framework enables formal comparison of seemingly different approaches. For instance, Proposition 2.10 demonstrates that energy minimization and autoencoding are equivalent under certain conditions - a relationship that's intuitive but hard to prove without our formal machinery. Similarly, Proposition 3.3 reveals manipulation as a refinement of CycleGAN, explaining why it achieves similar capabilities through simpler training. We do not know whether there exists a "systematic" way to relate descriptions of tasks to behaviours in the way you ask, and we aren't claiming we have such a method, because ML is complicated! However, part of our motivation for providing patterns alongside our proof examples was to give an accessible basis for beginning to investigate how these equational constraints inform model behaviour more generally.
>
> Regarding rules for diagram transformation: these are detailed in Appendix A, but we agree these could be better highlighted. Key operations include:
> - Task refinement (Definition 2.9)
> - Specialization (Section A.2)
> - Equational reasoning (exemplified in proofs B.12, B.14)
>
> Regarding comparative insights: We do provide several concrete examples, though these could be more explicitly signposted.
> - Proposition 2.10's equivalence explains that sufficiently powerful autoencoders successfully tackle energy minimization tasks (i.e. unsupervised learning) under appropriate conditions.
> - Proposition 3.3's refinement relationship predicts that manipulation would achieve CycleGAN-like capabilities, which we then verified experimentally.
> - In Section 3.3, understanding the formal requirements of manipulation helped us recognize that put could be implemented using an autoencoder and a learnt vector addition, while preserving desired behaviors.
>
>
> Addressing your specific questions:
>
> 1. The compound function α combines individual task losses into a single objective. As defined in Definition 2.2 (ln 111-124), α is a differentiable function (ℝ≥0)×|Φl| → ℝ≥0 that combines the individual divergences from atomic tasks. In practice, this is typically a weighted sum of the losses, as demonstrated in Appendix D.1's implementation where specific weights are given for each task component (ln 1487).
>
> 2. Yes, domain and codomain refer to inputs (left) and outputs (right) of diagrams respectively, as defined formally in Definition 2.1 (ln 94-97). For example, in Pattern 2.3's classification task, the domain is the data-label pair space and the codomain is the label space.
>
> 3. Precisely - as introduced on ln 54-55, "system" refers to the learned behavior being optimized and "specification" to the target behavior.
>
> 4. Good catch, thank you! Yes, there should be a Y in the domain to match the distribution 𝒴. We will correct this in the diagram.
>
> 5. The text above ∶⇒ indicates which learners' parameters are being optimized in that task. For PutPut, we're optimizing get's parameters to ensure it behaves consistently with the composition of puts. An explicit definition was found wanting by other reviewers too, so we will incorporate some version of the following into the paper: the symbol ∶⇒ denotes an abstract objective function to be minimised between the LHS and RHS diagrams (the formal semantics of which are given in Appendix A.1). This abstract objective becomes concrete modulo a choice of statistical divergence and weighting function in the context of other (atomic) tasks. If this is a helpful or clarifying framing, we will seek to include this explicitly.
>
> 6. To head off a potential misunderstanding, strong manipulation is regular manipulation plus additional tasks, so in this case a specialisation is immediate. More generally however, yes; this sort of schematised reasoning is exemplified in the proofs relating energy minimisation and autoencoder in Appendix B.1. B.12 and B.14 are pure diagrammatic reasoning.
>
> We see the framework's primary value as providing a mathematical "lingua franca" for reasoning about behavioral relationships across ML approaches. But, it's not a recipe book for implementation choices, and we cannot automagically derive equivalences between tasks. The framework wants to become a kind of toolkit to understand why certain approaches work or fail, and to predict relationships between seemingly different methods. We look forward to further discussions!

---

### Official Review · Reviewer_mqHa · 2024-11-05

**Soundness:** 3
**Presentation:** 1
**Contribution:** 1
**Rating:** 3
**Confidence:** 2

**Summary:**

The article describes a formal framework for defining objective functions: these are modeled as a weighted sum of several statistical divergence functions (e.g., log-likelihood, cross-entropy, etc.). Specifically, a diagrammatic language expressing equivalence between diagrams is used to model *atomic tasks*. In contrast, their linear combination (using user-provided scaling coefficients) is referred to as a *compound task* (or objective function).

To showcase the practical advantages of this formal language, the authors propose a style-transfer objective function obtained by borrowing concepts from the Bidirectional Transformation field.

**Strengths:**

The language provided could be used to succinctly express complex loss function, using a relatively intuitive and succinct grammar. The utilization of a standard notation for referring to minimization objectives could prove useful, as similar concepts have been adopted in computer science.
Furthermore, the formal nature of the grammar lends itself to using formal verification techniques to look for properties of interest in the objective function. How would be done in practice however I am still not sure.

**Weaknesses:**

- The article is sometimes difficult to understand and unpolished; this is particularly true for sections 1 and 2. They are difficult to read and the vocabulary is not always defined (e.g. what is a datatype, what does ∶⇒ mean? ). As it stands, the paper can be confusing to read, this might be due to unpolished writing and a foreign vocabulary from the one typically used in the ML community.

- The proposed language could be more intuitive, at least from a ML prospective; this may hinder its usage in the broader ML community.

- I am unsure if the benefits of this pattern language are a strong enough contribution for a publication at this venue. In general, the topic tackled by this work seems to not be a good fit for ICLR. A computer science conference or journal might be more pertinent.

- While I understand the necessity to abstract away the choice of architecture, training schedule, and similar hyper-parameters, more often than not they are crucial for the successful estimation of a ML model. The lack of any type of modeling for these components makes me doubtful of the actual practicality of this pattern language.

**Questions:**

1) How would the proposed language formalize size-varying data: For example, how would you represent the autoregressive cross-entropy loss commonly adopted by large language models?

2) What were actually the objective functions used during training for the various experiments? It would be nice to have a mapping from the diagrams shown in Task 3.1 to a conventional mathematically expressed objective function.

3) How were the task weights reported in Appendix D estimated? Were they chosen arbitrarily or a hyperparameter search was performed?

4) What does the text above the ∶⇒ symbol mean? Does it indicate the parameters optimized by the loss function?

5) What does the ∶⇒ symbol formally mean? Is there a definition in the appendix that I have missed? Is it a bidirectional relation?

---

> ### Author Response · Authors · 2024-11-17
>
> We thank you for your thoughtful and detailed feedback. We particularly value your attention to formal clarity and practical ML concerns. We note that several of your questions point to areas where we can improve the presentation to better serve the ML community, which we address below.
>
> Before addressing your questions in order, we understand your crucial concerns to be:
> 1. How the formalism handles practical ML concerns
> 2. Need for precise mathematical definitions
> 3. Appropriateness for an ML venue
>
> 1. Regarding practical ML concerns, our framework explicitly accommodates architectural choices and variable-length data. As demonstrated in Section 3.3 (ln 380-387), practitioners can freely choose architectures - we showed how restricting put to be linear yielded interpretable latent spaces without compromising the formal guarantees. For variable-length data like text, we note that our framework operates at the level of abstract datatypes (detailed in Appendix A.1); concrete implementations handle variable-length data through standard approaches like padding or autoregressive processing, as exemplified in Appendix F.1's text experiments.
>
> 2. Concerning mathematical precision, we acknowledge that some notation (particularly ∶⇒) needed better introduction. The symbol ∶⇒ denotes an abstract objective function to be minimised between the LHS and RHS diagrams (the formal semantics of which are given in Appendix A.1). This abstract objective becomes concrete modulo a choice of statistical divergence and weighting function in the context of other (atomic) tasks. If this is a helpful or clarifying framing, we will seek to include this explicitly.
>
> 3. Regarding venue fit, while our framework draws from category theory, its primary contribution is to ML practice through:
> - Enabling systematic design of objective functions (Section 3)
> - Providing formal guarantees about learned behaviour (e.g. Props 3.3, 3.4)
> - Guiding architecture choices (Section 3.3's linear put)
> - Facilitating discovery of new training approaches (manipulation task)
>
> The core systematic recipe of the paper is *learning representations of equational theories*, which we will include explicitly in the paper if you find the perspective helpful:
> 1. Begin with an equational specification of desired behaviour - formal constraints that captures how components should interact. While there's no universal recipe for converting desired behaviors into equations, we provide a foundation of well-understood patterns (Section 2) that can be analyzed and composed to derive novel objectives - as demonstrated by our manipulation task. As novel examples, Section 3.1 specifies what properties a well-behaved attribute editor should satisfy, and Appendix E specifies how to translate algebraic conditions governing push and pop into a neural stack.
> 2. Replace abstract functions with learnable components (e.g., neural networks). The framework is agnostic to specific architectural choices, treating learners as universal approximators (detailed in Appendix A.2).
> 3. Transform equations into trainable objectives by choosing appropriate statistical divergences and combining weights (subject to hyperparameter search). These choices determine the specific overall loss function to be minimised. The resulting trained components will exhibit the specified behavior (in the computational limit) because the learning process finds smooth function approximations satisfying the constraints.
>
>
> Addressing specific questions:
>
> "How would the proposed language formalize size-varying data?"
> As noted above, our framework operates at the level of abstract datatypes (Appendix A.1). Implementation details for variable-length data are demonstrated in Appendix F.1's text experiments.
>
> "What were the objective functions used during training?"
> The complete objective functions with weights are detailed in Appendix D. For example, D.1 provides the full weighted combination of losses for the Spriteworld experiments (ln 1476-1484).
>
> "How were task weights estimated?"
> As detailed in Appendix D, weights were chosen through standard ML practice (hyperparameter search) to balance the different loss terms. We acknowledge this could be better explained.
>
> "What does the text above ∶⇒ mean?"
> The text indicates which learnable parameters are being optimised in that task, as explained in Definition 2.1 (ln 94-97).
>
> "What does ∶⇒ formally mean?"
> As noted above, ∶⇒ denotes an abstract objective function minimising some divergence between the LHS and RHS diagrams, to be made concrete by supplying a choice of divergence and weighting function.
>
> We appreciate your careful attention to formal precision. While maintaining mathematical rigor, we propose adding clearer introductions to notation and better signposting of definitions to improve accessibility for the ML community. We welcome further discussion on any aspects requiring clarification.

---

### Official Review · Reviewer_2Po6 · 2024-11-06

**Soundness:** 3
**Presentation:** 1
**Contribution:** 2
**Rating:** 3
**Confidence:** 2

**Summary:**

This work introduces a diagram-based language for creating and manipulating ML models with the notion of the task as the fundamental unit of computation. Several types of common models are given in terms of this framework such as autoencoders, classifiers, and in general, manipulators. Experiments are performed on manipulation (attribute selection and editing) for MNIST, a sprites-like dataset of shapes, and CelabA. it seems that the proposed method effectively can perform disentanglement and attribute editing to some degree.

----------------------------------------------------------------------------
I believe the authors provided a good faith effort to address concerns raised here. Therefore I will update my score(s) accordingly.

**Strengths:**

Originality: I think there is novelty in the category-theoretic foundations of this framework. I haven't seen this diagram notation before.

Clarity: I think the writing was easy to follow, my concerns below notwithstanding.

Significance: its clearly very early work, possibly a more developed version of this work could be impactful.

**Weaknesses:**

I found this paper extremely difficult to parse for content. It seems like there is something here, but it is hardly even hinted at in the main paper. I did not thoroughly check the proofs in the appendix so possibly I missed some important details there. But in general I fail to see what is actually being proposed here. My concerns are as follows:

* Over half of the main paper is spent restating some form of empirical risk minimization, and reintroducing commonly used notation and modeling paradigms like autoencoders and classifiers. In this exposition I didn't find anything new except different jargon e.g. the operands of the divergence measure called sys and spec.

* I'm unsure of what was even done for the experiments. Was something implemented? Explaining how this framework gets turned into an algorithm or implemented would be useful. Right now all we have is a small diagram and then results which is impossible to interpret.

* Much of the nomenclature used here like get, putput, fake, fool, undoability is confusing at best. I don't see how this particular exposition is useful or gets the reader any closer to understanding the framework.

* I found 3.2 to be a very obscure experiment and dataset. Can we do anything else but attribute editing (manipulation)?

**Questions:**

* In what sense are the given CycleGAN generators optimal? Is this optimality with respect to style transfer in general or just CycleGAN?

* What role do these diagrams play? I don't find them in any way intuitive if they are purely illustrative.

* What is Undoability? I can't parse the diagram well enough to be sure.

* How does this framework generate new or better components / tasks; are there any examples of improvements over the standard approach?

* Why is attribute editing chosen as the only task for this work? The first sections of the paper belabor the point that all kinds of modeling decisions can be expressed in this framework. So why don't we have autoencoders and classifier examples?

---

> ### Author Response · Authors · 2024-11-17
>
> We thank you for your constructive and challenging feedback. We particularly appreciate your careful attention to practical concerns, which we agree are crucial for bridging theoretical frameworks to machine learning practice. However, we note that several of the concerns you raise are addressed in the paper and appendices, which we detail below.
>
> Before we address all of your questions in order, we understand your crucial concerns to be:
> A. How theory translates to concrete implementations
> B. Practical or conceptual advantages over standard approaches
> C. Justification for focusing on attribute editing given the framework's generality
>
> A. Concerning implementation, Appendix D provides comprehensive details of architectures, hyperparameters, and training procedures. We are also happy to answer any further methodological questions you have! While there are signposts to these appendices in each substantial Figure displaying results (ln 298, ln 375, ln 429), we propose adding a subsection to Section 3 explicitly showing how diagrammatic specifications translate to familiar ML objectives and architectures, and more explicitly directing the reader to the experimental details.
>
> B. Concerning conceptual advantages, it is true that regularisation and multi-task learning are expressible in terms of tasks, but in addition, we may express "multi-learner multi-task learning", which does not appear to be a named concept; in this sense 'task' is a unifying view. Concerning subsequent practical advantages of this perspective, our framework provides a systematic recipe for developing new ML training objectives, which we will include explicitly in the paper if you find the perspective helpful:
> 1. Begin with an equational specification of desired behaviour - formal constraints that captures how components should interact. While there's no universal recipe for converting desired behaviors into equations, we provide a foundation of well-understood patterns (Section 2) that can be analyzed and composed to derive novel objectives - as demonstrated by our manipulation task. As novel examples, Section 3.1 specifies what properties a well-behaved attribute editor should satisfy, and Appendix E specifies how to translate algebraic conditions governing push and pop into a neural stack.
> 2. Replace abstract functions with learnable components (e.g., neural networks). The framework is agnostic to specific architectural choices, treating learners as universal approximators (detailed in Appendix A.2).
> 3. Transform equations into trainable objectives by choosing appropriate statistical divergences and combining weights (subject to hyperparameter search). These choices determine the specific overall loss function to be minimised. The resulting trained components will exhibit the specified behavior (in the computational limit) because the learning process finds smooth function approximations satisfying the constraints. As detailed in our answer to "How does this framework generate new or better components/tasks?", this recipe does deliver practical advantages.
>
> C. Regarding the focus on attribute editing: manipulation serves as a compelling proof-of-concept precisely because it demonstrates deriving novel training approaches from formal specifications. Additionally, Appendix E shows broader applicability through "neural data structures", demonstrating how our pattern language guides implementation of differentiable versions of classical data structures. We elaborate further in the answer to "Why is attribute editing chosen as the only task?".
>
> (comment continues)

---

> > ### Author Response · Authors · 2024-11-17
> >
> > Addressing specific questions:
> >
> > "In what sense are the given CycleGAN generators optimal?"
> >
> > Our apologies, this is a definition we have left to the Appendix where the proposition is proved, which we can signpost better. In the case of cycleGAN and strong manipulator where there are adversarial objectives between learners, it is a priori impossible for there to exist learnt functions that 'perfectly' satisfy all tasks. 'Optimality' is our attempt to define the next best thing from 'perfect', glossed as 'when the learnt functions satisfy all tasks as well as possible', and viewable as an equilibrium between learners with adversarial loss functions.
> >
> >
> > "What role do these diagrams play?"
> >
> > Purely formally, they provide crucial advantages over traditional notation. First, they encode typing information for safe composition of learners, which is typically obscured in standard mathematical notation. For example, manipulator involves two learners and two datatypes, and it is necessary to disambiguate the putget and getput composites. Second, they handle multi-output operations naturally (e.g. complement manipulator in Task 3.5 and stack examples in Appendix E), which are cumbersome in standard infix notation, requiring indices and some symbolic redundancy.
> >
> > In term of accessibility and theoretical benefits, at their simplest, the diagrams can be read as familiar flowcharts showing how data and learners interact. They enable rigorous reasoning about relationships between tasks, as demonstrated in Propositions 2.10 and 3.3. They support a visual style of proof (exemplified in proofs of Lemmas B.12 and B.14) that makes formal arguments more tractable. They guide implementation by making clear which components need to communicate and how. The framework's ability to both prove theoretical relationships (like the equivalence of energy minimisation and autoencoding in Proposition 2.10) and guide practical implementations (like the manipulation task) further demonstrates their utility beyond mere notation.
> >
> >
> > "What is Undoability?"
> >
> > Undoability specifies that attribute edits can be reversed. As signposted in line 254, the conceptual explanation of these objectives in prose are in Appendix C.1, where (line 1373), Undoability in the context of shape-colour is the requirement that "editing again with a second put to restore the read-out colour of the original shape must restore the original image."
> >
> >
> > "How does this framework generate new or better components/tasks?"
> >
> > We agree wholeheartedly that it is important to demonstrate the practical value of theory. Concerning concrete practical advantages over standard approaches, our modular framework enables the following. As stated in 3.1 (line 276) we have a novel class- and style- conditioned generative model that edits attributes:
> > - deterministically (cf. VAEs)
> > - with non-adversarial training (cf. GANs)
> > - without interventions on the data or hardcoding features in latent spaces (cf. CVAEs)
> > - and learnt without supervised pairs
> > Moreover, we have demonstrated the ability to modularly improve components: in Section 3.3, we are able to restrict the put to be a linear modulation on a latent space, which separates representations of smiling and non-smiling faces, and obtains "smile" and "no-smile" concept vectors; all these are bonuses obtained without compromising the put's role as an attribute editor.
> >
> >
> > "Why is attribute editing chosen as the only task?"
> >
> > Throughout, we treat manipulation as a novel running example or case study that demonstrates our theoretical framework. As you have noted, it is not the only task: section 2 does reformulate common ML approaches as tasks, including generic supervised learning (Pattern 2.3 and footnote 1), two variants of autoencoding (Patts. 2.4 & 2.7), two GAN variants (Patts. 2.5 & 2.7), and a broad class of unsupervised learning approaches including PCA and k-means (Patt. 2.8 & ln. 187). Beyond what is already known, as signposted in line 517 of the conclusion, Appendix E sketches further "novel" tasks detailing how a modular approach to learners implements neural data structures. We note a potential misunderstanding in the phrasing of this question, that the framework of tasks expresses modelling decisions. Interpreting modelling decisions to mean things like choices of architecture and learning rates, tasks abstract away such choices, which leaves them to be implemented by the practitioner according to experience. As section 3.3 gives direct evidence for, these kinds of choices do not compromise the basic specification of desired behaviour via tasks.
> >
> >
> >
> > Given these clarifications and the evidence presented in the paper, we believe the practical value and implementation details are substantially stronger than suggested by your current assessment. We welcome further discussion on any specific aspects that remain unclear or concerning, and we look forward to integrating clarifying perspectives into the text.

---

> > > ### Comment · Reviewer_2Po6 · 2024-11-22
> > > **Response to Rebuttal**
> > >
> > > Thank you for providing such a detailed response, I will consolidate my reply here, addressing each point in order.
> > > To start, I believe that the work presented here is interesting, but not communicated well to an ML audience.
> > >
> > > A) I did see implementation details in the appendix, including a description of hyperparameters and architectural choices. What doesn't follow from the current draft is how we go from the diagram description of a task or problem, to an implementation with that given selection of hyperparameters and architecture decisions. I don't think this is just a practical consideration, but it points to the ability of the framework to generate a testable system. My impression is that this framework takes us from a behavioral description to a high level architecture, from which we perform HP+arch search. But this is already what the field does.
> > >
> > > B) As far as communicating the advantages of this framework, it would be very helpful to lead with a compelling example. If "a systematic recipe for developing new ML training objectives" is such an example then it would be quite useful. Taking the manipulation task (sec 3) as an example, its too difficult in the current draft to understand what is interesting there. Much of the nomenclature is a remapping of words we already have (which is fine) but it makes it difficult to pick out what new terms or concepts are worth going back and forth to understand.
> > >
> > > **Diagrams**: I actually like the diagram approach as an alternative / complement to complex notation. I also believe it can be presented better than the current draft.
> > >
> > > **Undoability**: I could have been more clear, my question was about whether or not undoability (and other components like putput) are fundamental in any sense, or are they just labels attached to the particular diagrams generated for this problem. Components like classify appear general, but undoability does not. This points at a broader question. Does this paper propose a fundamental set of operations / diagrams that can be composed into learners? The top of section 2 seems to imply that there is such a fundamental set.
> > >
> > > **Generating new components**: I agree that the proposed manipulation method is an example, but I was curious how the authors see their framework contrasted with the standard operating procedure. It is standard to start with some mathematical description of desired behavior, instantiate modeling decisions as an architecture, and then implement. I think it would be helpful to see how this framework more naturally leads to better modeling decisions. For example, an idea would be to take the VAE [1] formulation, and walk through its instantiation as a architecture, then do the same for this framework, showing at each step the advantages of the proposed method.
> > >
> > > **Tasks**: "the framework of tasks expresses modelling decisions. Interpreting modelling decisions to mean things like choices of architecture and learning rates, tasks abstract away such choices, which leaves them to be implemented by the practitioner according to experience".
> > >
> > > * This is an important point that could be made more clearly in the draft, possibly in lines 54-74. There is this: "We abstract away implementation details such as architecture and training by idealising [...]", but I think this does not make it clear enough that architecture and training are _not meant_ to follow from this framework.
> > >
> > > --------------------------------------------------
> > >
> > > Overall, I appreciate that the authors provided a thorough response. Pending the rest of this discussion period I am at least amenable to raising my score to a 4/5. I think that the draft in its current form needs significant work on communication to the ICLR audience, or is not a good fit for ICLR.
> > >
> > > I noted that the authors are hesitant to move too many details to the main text. I would reconsider this if they mean to keep targeting the ICLR audience. I believe it would go a long way to have some implementation details described in section 3, while noting that these details are up to the practitioner. The authors have already done the work of providing/finding these details, so it cuts the story short to leave those all the way down in appendix sec. D.
> > >
> > >
> > > [1] Kingma, Diederik P. "Auto-encoding variational bayes." arXiv preprint arXiv:1312.6114 (2013).

---

> > > > ### Author Response · Authors · 2024-11-23
> > > > **2nd Rebuttal, Pt 1**
> > > >
> > > > > What doesn't follow from the current draft is how we go from the diagram description of a task or problem, to an implementation with that given selection of hyperparameters and architecture decisions... I believe it would go a long way to have some implementation details described in section 3, while noting that these details are up to the practitioner.
> > > >
> > > > Thank you for pointing out our blindspot and following up with this constructive suggestion. We will do this, as well as integrate the clarification you suggest in lines 54-74.
> > > >
> > > > > My impression is that this framework takes us from a behavioral description to a high level architecture, from which we perform HP+arch search. But this is already what the field does.
> > > >
> > > > Yes; that is exactly what we do already. The question of course is what a "behavioural description" is formally, and we answer that they often take the form of pseudo-equations. As far as we know, this is the first time someone has bothered to write down an answer, perhaps because the question itself is not a pressing or practical one. This leads us naturally to address what we suspect may be the deepest concern.
> > > >
> > > > > I believe that the work presented here is interesting, but not communicated well to an ML audience... I think that the draft in its current form needs significant work on communication to the ICLR audience, or is not a good fit for ICLR.
> > > >
> > > > We consider these remarks to be crucial! Please allow us to steelman this observation before addressing it; we don't mean to put words in your mouth, but we will recount a common type of objection we are familiar with, as we have an ugly-duckling problem of being too applied for pure theory, and too theoretical for applied. What follows is the strongest sort of objection from a typical engineer, which perhaps encapsulates the topic-venue mismatch you allude to.
> > > >
> > > > "Of course, everyone in the ML community already knows *how to* turn behavioural specifications *regardless of what they are formally* into architectures in the way you describe. Sure it might be *interesting*, but what exactly is the practical value to be gained from this different perspective? Less charitably, why waste time with formal effort just to *say what everyone already knows?*"
> > > >
> > > > Our answer is that providing a framework to answer "what is a behavioural specification", we are dealing with a "what is" sort of question, opposed to a "how to" sort of question: to put it bluntly, we are trying to be scientists among craftsmen. Of course nobody is seriously against the abstract idea of the long-term value of science, but what good is it in this particular instance? We will now provide our argument in three points, with recourse to evidence provided in this set of reviews.
> > > >
> > > > **1) Of course we need the right vocabulary.** As you contend: "Much of the nomenclature is a remapping of words we already have", but what if the words you have make unnecessary distinctions, or they fail to make necessary distinctions? Indeed we find this is so, and we will give one example in the text, and one example with reference to the reviews.
> > > >
> > > > **1.1)** Here is a simple but fundamental example of an unnecessary distinction. In our Pattern 2.3, we remark that what we call "classification" encompasses "regression"; the usual elementary ML distinction between discrete and continuous supervised learning amounts to an implementation detail (are the systems in question discrete or continuous). We might have named the pattern "supervised learning" instead.
> > > >
> > > > **1.2)** Here is another example of an unnecessary distinction, arising from a confusion our colleague GYtH has: "it's unclear to me how "(composite) tasks" differ from multitask learning or adding regularization terms." On our view, additive terms in an objective function correspond to tasks, so the distinction is artificial: mathematically, a regularisation term is just another task/constraint placed on the learner. Of course, if one insists that the distinction is meaningful, how does one distinguish between task-terms and regularisation-terms apart from the private imagination of the modeller? We provide an answer for this in 2).
> > > >
> > > > **1.3)** Here is an example of a natural consequence of our perspective that, to the best of our knowledge, the ML community does not have the words for. The closest gloss might be "multi-learner multi-objective learning", which reflects the predominant 'cultural default' in ML of training a single architecture to solve a problem. We elaborate more on this in 3).

---

> > > > > ### Author Response · Authors · 2024-11-23
> > > > > **2nd Rebuttal, pt. 2**
> > > > >
> > > > > **2) We can use the same terminology and intuitions of the field to explain things more easily; our framework permits us to actually use the terminology of the field as explanations as opposed to mere descriptions.** For example, you suggest that we walk through the VAE by Kingma and Welling; we in fact already dedicate all of lns 162-179 and Task 2.7 to this exact case study! The classic probabilistic reasoning in Kingma ("why it works") is opaque compared to the gloss we would like to use as engineers ("how it works"): the encoder embeds into a (mean,variance) space of gaussians, the decoder samples randomly from the gaussian with such a mean-variance pair before decoding, and there is a normalising regularisation term that asks encoders to tend towards unit gaussians. We did not think it necessary to write down our thinking in detail for this case study, but we will expand it here and seek to include anything you may find interesting.
> > > > >
> > > > > If we know how objective function terms correspond to behaviours, we aren't solely reliant on e.g. the probabilistic reasoning of Kingma and Welling to understand why a VAE works. Kingma and Welling themselves explain the VAE in similar prosaic terms on pg4 for the estimator before their reparameterisation: there's a reconstruction term, and then there's a regularisation term. The former makes the architecture behave like an autoencoder, and the latter makes encoded representations like gaussians, which means easy sampling and continuous interpolation. This second kind of explanation is fast and intuitive; certainly easier than reading all of K&W. We contend that such explanations are moreover *formally admissable*. There are basically two questions to answer to make such an explanation formal, both regarding (cf. 1.2) the correspondence between the 'words we use' and mathematical definitions. These are:
> > > > >
> > > > > **2.1)** What is a reconstruction loss term, and how do we know that we are looking at one; i.e. why are we allowed to call the VAE an autoencoder? On our account, this is because the 'reconstruction loss' of the VAE is a specialisation of the usual reconstruction loss of a regular autoencoder, and the proof of this is the dashed box in the left of Task 2.7, which exhibits the replacement of a deterministic decoder with a specialised sample-then-decode. The formal semantics of such specialisations are given in Appendix A.2.
> > > > >
> > > > > **2.2)** Do we have enough information in this presentation to conclude that the other term is a "regularisation term"; i.e. why do we call this term "normalise"? Yes. The satisfaction of reconstruction already yields an autoencoder, so this term can only be providing an additional constraint, and it suffices to demonstrate that this constraint is nontrivial. For this, consider that without the regularisation term, any regular autoencoder automatially yields a VAE by embedding into the subspace of gaussians with variance 0, yielding degenerate deterministic VAEs.

---

> > > > > > ### Author Response · Authors · 2024-11-23
> > > > > > **2nd Rebuttal, pt 3**
> > > > > >
> > > > > > **3) On the value of mere perspectives.**
> > > > > >
> > > > > > **3.1) Bridging fields.**
> > > > > >
> > > > > > > It is [standard operating procedure] to start with some mathematical description of desired behavior, instantiate modeling decisions as an architecture, and then implement.
> > > > > >
> > > > > > As stated in lns 276-278, the extant "standard procedure" ways to edit the attributes of data have limitations, where the choice is between adversarial and unstable training conditions with few guarantees (e.g. cycleGAN), or hardcoding latents and inductive biases: there is no other "pure" end-to-end approach that we know of like ours, because stating the editing task as a supervised objective for a single learner can't be done without before-and-after data pairs, which aren't available here. As far as we can tell, before our work, there was no pure end-to-end approach that actually followed the standard procedure you describe: take the mathematical description of the desired behaviour (e.g. lenses from database theory characterise editing attributes of data) and then implement. Lenses are not new, and they are very elementary constructs of applied algebra, and ML is a large field: if this standard procedure existed, it's hard to believe that such low hanging fruit had to wait until now. We contend that the "standard procedure" is an informal guidelines, and that our bridge is a formal variant of this procedure that bridges between higher-algebra and ML.
> > > > > >
> > > > > > **3.2) What's hard for you is easy for me, and vice versa.**
> > > > > >
> > > > > > > Taking the manipulation task (sec 3) as an example, its too difficult in the current draft to understand what is interesting there.
> > > > > >
> > > > > > Hopefully the task itself is clear as a style- and input- conditioned generation task: given a data *d* and an attribute *a*, make the minimal edit to the data so that it has the attribute, e.g. change the colour of a shape, make a person smile, or change a numeral without changing handwriting style. If the task is clear but our method of solution is not (even after referring to Appendix C.1 for prosaic descriptions of each subtask of manipulator), then we will try an alternative explanation:
> > > > > >
> > > > > > Our approach uses multiple learners that *mutually define one another*, and as we note in our response to GYtH, this kind of thinking isn't totally alien to ML: in the game-theoretic explanation of a GAN, the generator seeks to FOOL a discriminator, and the discriminator seeks to detect the FAKEs from a generator (and this is why we name the tasks so). As both gen and dsc mutually improve by competition, eventually the generator may surpass a human discriminator, at which point we discard the discriminator and present gen alone as a generative model.
> > > > > >
> > > > > > In a manipulator, the put and get are not in competition, but they must cooperate to satisfy certain tasks. In the same way that FAKE and FOOL yield an explanation of why a GAN works in terms of agents and their interactions, the various PutPut and Undoability and other tasks (explained in C.1) govern the interaction of get and put, allowing them to mutually define one another.

---

> > > > > > > ### Author Response · Authors · 2024-11-23
> > > > > > > **2nd Rebuttal, pt 5**
> > > > > > >
> > > > > > > **On the limits of formalism.**
> > > > > > >
> > > > > > > > are [tasks] fundamental in any sense, or are they just labels attached to the particular diagrams generated for this problem. Components like classify appear general, but undoability does not. This points at a broader question. Does this paper propose a fundamental set of operations / diagrams that can be composed into learners?
> > > > > > >
> > > > > > > Put another way, how can we tell which tasks are general and important and which are not, and does there exist a fundamental set of tasks we can compose to do everything? This is *the* question, and our answer is in the title!
> > > > > > >
> > > > > > > No, we do not know of a basic set of tasks, and we do not know how to, in full generality, relate an arbitrary behavioural specific in natural language to a formal objective function. If such a totalising systematic framework existed, then ML engineering would purely consist of, as you say, "HP+arch search", which would be an insult to the ML community.
> > > > > > >
> > > > > > > However, what we do know from practical and collective experience in the field of ML is how various learning paradigms behave; we can recognise classification, autoencoding, adversarial training and normalisation terms, and so on. These are like patterns in software engineering, or in architecture: there is no one correct way to write a program or construct a building, but there are best practices and a stock of basic building blocks that are often used to compose larger solutions. If the nature of ML as a whole is similar to classical architecture (between art and engineering), then asking for a fundamental set of tasks is like asking an architect for a fundamental set of design primitives and materials; a poorly-typed question.
> > > > > > >
> > > > > > > We argued that some tasks deserve the name "pattern" not just because they are common and well-studied, but also because we can make sense of other tasks in terms of patterns. As we note to reviewer zN4b, using task refinement (Definition 2.9), specialization (Section A.2), and equational reasoning (exemplified in proofs B.12, B.14), we can reason to extend our vocabulary of correspondences between tasks and behaviours. So you ask what the difference may be between a "pattern" like classification, and a 'mere task' like undoability: as in any language, the difference is only how we feel about it. Before GANs, adversarial training conditions were not a pattern, and now they arguably are patterns. Perhaps one day 'mere tasks' like manipulator and undoability will be well-understood and broadly used enough to become patterns.
> > > > > > >
> > > > > > > In our view, this language of patterns is pretty close to how the ML community already thinks about things, and all we are doing is making it formal and trying to take it seriously: we think we must be doing a good job on this count, because you noted in your original review that we seemed to be spending a lot of time talking about things that everyone already knows!

---

> > > > > > > > ### Author Response · Authors · 2024-11-23
> > > > > > > > **2nd Rebuttal: Concluding questions and concerns.**
> > > > > > > >
> > > > > > > > **Our questions to you, and our concerns**
> > > > > > > >
> > > > > > > > **C1: Difficult or uninteresting?** You noted before that the task is "obscure". Perhaps this is the case because it's not obvious how to spell out a characterising objective for manipulation when restricted to a single architecture, without recourse to inductive biases, hardcoding, or supervised before-and-after pairs; this conclusion is evidentially supported by the extant literature, unless you know of any counterexamples. We have a formal framework that interprets algebraic specifications as multi-learner multi-objective learning that demonstrably works in a novel way; *is this not easy to understand, or is it uninteresting?* If it is a matter of difficulty, we are happy to continue clarifying any technical points of difficult that may arise. If it is a matter of interest (though you do find the work interesting), *to what extent are your value judgements indexed on practical/procedural contributions over theoretical and conceptual contributions?*
> > > > > > > >
> > > > > > > > **C2: Inconsistencies.** We appreciate your willingness to raise your overall score, but we are concerned about inconsistencies in your two reviews; you first claimed our writing was "easy to follow" yet "extremely difficult to parse for content"; you initially asked "Was something implemented?" then later acknowledged seeing the implementation details in the appendix; you first found diagrams "not in any way intuitive" but later "like the diagram approach"; you claimed to find "nothing new except different jargon" yet later acknowledge the work as "interesting"; you described our experiments as "obscure" yet later demonstrate understanding while claiming they're "too difficult to understand what is interesting".
> > > > > > > >
> > > > > > > > We find it difficult to account for these discrepancies by the effect of the content of our response to you, because all of our substantial arguments are perspectival rather than introducing new technical content beyond material that is already explicit or implicit in the text. The alternative explanation, and our concern, is that you have only engaged shallowly with the technical and conceptual content of our work. To dispel our concerns and for the record, could you please answer the following?
> > > > > > > >
> > > > > > > > *Can you please point out specific technical or mathematical errors that justify your soundness score of 1?*
> > > > > > > >
> > > > > > > > *Can you please point out superceding prior work that justifies your contribution score of 1?*
> > > > > > > >
> > > > > > > > **C3: "Significant work" already present**
> > > > > > > >
> > > > > > > > Your responses consistently request material that is already present in the original submission. You suggest walking through the VAE case study "showing at each step the advantages of the proposed method" - this is precisely what Task 2.7 and lines 162-179 provide, detailing the specialization of decoder and normalization constraints. You ask for implementation details in Section 3, yet later acknowledge finding these in Appendix D. You request clarification on how "we go from diagram description to implementation" when Section 3 demonstrates this end-to-end for the manipulation task. You seek understanding of whether components like undoability are fundamental, yet Section C.1 provides detailed prosaic explanations of each task's purpose. You ask for better explanation of how our framework improves modeling decisions, yet Section 3 provides a complete case study deriving a novel training approach with concrete advantages.
> > > > > > > >
> > > > > > > > This pattern of requesting what is already present, combined with the inconsistencies noted in C2 and your admission of not thoroughly checking the appendix proofs while assigning a soundness score of 1, suggests your assessment may be based on a shallow reading of our work. If we have misunderstood your position, we welcome specific technical criticisms or identification of gaps in our framework's presentation. Otherwise, to dispel our concerns:
> > > > > > > >
> > > > > > > > *Would you agree that your assessment would be more appropriately reflected with a lower confidence score, given these observations?*

---

### Author Response · Authors · 2024-11-28
**Revision changelog (1/2)**

We would like to thank the reviewers for their thoughtful feedback, which we have taken to heart. Our first submission did not adequately explain key aspects of our work clearly, and for this we apologise. In our submitted revision, we have made major rewrites and restructuring that we hope will more clearly communicate our ideas: all of our revisions are a direct result of your questions and discussion, and we thank you for helping us make this work stronger. For the reviewers' convenience, all major changes in the text have been coloured blue.

**Changelog**

The abstract has been rephrased to better reflect the focal points of our paper as we see it.

The introduction has been restructured in the following ways:

- The introduction is now centred around developing *autoencoder* as a running example.
- Our introductory figures now include a direct translation between our diagrammatic notation and objective functions
- We clarify the diagrammatic technique we use of "specialisation" as encompassing architectural choices and multi-objective learning. As examples, we provide a worked example of adding residuals, and perceptual losses, respectively.
- Moving an example upwards, we demonstrate immediately how specialisation may be used to obtain VAEs from autoencoders.
- We have streamlined the presentation to clarify the role and value of string-diagrammatic notation over standard symbolic notation.
- We have added a contributions subsection that clearly outlines our major contributions, and should better indicate that we consider the nature of our contribution to be primarily theoretical, where our practical component is primarily a verification or proof-of-concept.
1. (Theory) The formalisation of the standard procedure in ML
2. (Theory) The ability to analyse, design, and predict behaviour of models
3. (Practical) the manipulation task as a proof-of-concept for the value of the theory
- The conceptual novelty of our approach to manipulation has been highlighted; it is to the best of our knowledge the first instance of a problem class naturally tackled by a relatively underexplored "multi-learner multi-objective" approach to learning.

The "Tasks and Patterns" section has been updated in the following ways:

- Task notation is now properly introduced, and task notation throughout has been changed to double-harpoons.
- The way by which tasks relate to concrete objective functions is now explicit.
- The motivation and value of "patterns" is elaborated: in the absence of a systematic method to translate natural-language behaviour specs to tasks, we instead provide design patterns as a provisional "basic vocabulary"
- Reasoning about behaviour using tasks is now highlighted with respect to the VAE example

(continued)

---

> ### Author Response · Authors · 2024-11-28
> **Revision changelog (2/2)**
>
> The "Experiment" section has been updated in the following ways:
>
> - Manipulation is now presented alongside objective functions for each task
> - "Strong manipulation" was a source of confusion, and it has been moved to the appendix, as it is primarily a complex regularisation term that we only use for theoretical proofs rather than our practical experiments.
> - The prose intuitions for manipulation have been moved up from the appendix; now they immediately explain what each task does
> - A full presentation of the complete manipulation task in standard notation as a hyperparameterised-objective-function has been added; we hope that it serves as a vivid demonstration of the relative informational compactness of string diagrams!
> - The two propositions that tell us something about the behaviour of manipulation have been rephrased in a more accessible and hopefully intuitive manner
> - There is now an explicit signpost that methodologies are in the appendix and only results are reported; putting the methodologies in the main body would put us badly over the page limit, and they provide relatively little conceptual value, since they are mostly reporting hyperparameters
> - The linear put has been explicitly signposted as an instance of specialisation, and the figure has changed to reflect this
> - Details of interest have been highlighted in Figure 6 reporting the faces-experiment: the two remarkable findings are that linear put allows continuous interpolation even when only discrete labels are provided in the dataset, and that we observe a separation of class-representations in latent space without an explicit regularisation term that aims to achieve it.
> - There are non-critical spacing issues on page 8 that we will address in a future camera-ready version, pending more revisions.
>
> The Appendices have been updated in the following ways:
>
> - Appendix A is now a self-contained exposition of string diagrams, their syntax and semantics. In particular, the figure listing the stock of generators has been moved here; as we emphasise in the introduction, the point of the string diagrams is that they match up with readily available flowchart-intuitions, so there was perhaps little conceptual value added in having this figure in the main body.
> - Strong manipulation and its role and effect as a regularisation term have been collected under Appendix B
>
>
> We were concerned that in the previous presentation, due to shortcomings in our presentation, judgement of our work was overweighted on the basis of a practical contribution that we openly admit is limited to toy domains: it was never our intention to claim that manipulation is objectively impressive! We hope that these revisions better reflect our intent to communicate our contribution primarily as a new theoretical framework, for which we have done our due diligence by verifying its predictions and potential benefits practically in the context of a novel task we design, implement, and analyse.
>
> We look forward to continuing discussions on the basis of these much-needed changes, and we once again thank the reviewers for their time and care.

---

### Meta-Review · Area_Chair_7281 · 2024-12-23

**Metareview:**

All reviewers seemed to recognise some merit in the author's attempt at a pattern language, with comments suggesting this may aid conceptual understanding and newcomers to the field and help relate various algorithms among other possible use cases. There were also some positive comments about the novelty of the "category-theoretic foundations of this framework".

Unfortunately, a strong common criticism of this submission is a lack of clarity, both in presentation (redefinition of known terms, undefined symbols, unclear transition from framework to experiments) and the overall goal of the submission (it being unclear how this can help advance theoretical/mathematical understanding or improve empirical results). There was some concern by several reviewers that the ICLR audience may not be the correct audience for this submission.

As a result and due to the lack of a strong proponent of this work among the reviewers, I'm afraid this submission is not currently suitable for publication at ICLR.

**Additional Comments On Reviewer Discussion:**

I would have liked to see a livelier discussion between reviewers and authors, with reviewers (zN4b, mqHa) unfortunately not responding to the authors' detailed response. Nevertheless, we saw a healthy exchange between authors and reviewers (2Po6, GYtH), although it looks like no score update resulted from these discussions despite reviewer 2Po6 suggesting this may be possible: "I am at least amenable to raising my score to a 4/5".

---

### Decision · Program_Chairs · 2025-01-22

Reject